

**Growth of the coccolithophore *Emiliania huxleyi* in light- and nutrient-limited batch**
**reactors: relevance for the BIOSOPE deep ecological niche of coccolithophores**
**L. Perrin[1], I. Probert[2], G. Langer[3] and G. Aloisi[4]**
[1]Sorbonne Universités, UPMC Univ. Paris 06 -CNRS-IRD-MNHN, LOCEAN-IPSL, 75252 Paris, France.
[2]CNRS-UPMC Univ. Paris 06  FR2424, Roscoff Culture Collection, Station Biologique de Roscoff, 29680
Roscoff, France.
[3]Marine Biological Association, The Laboratory, Citadel Hill, Plymouth PL1 2PB, UK.
[4]LOCEAN, UMR 7159, CNRS-UPMC-IRD-MNHN, 75252 Paris, France.
Correspondence to:  L. Perrin (lpelod@locean-ipsl.upmc.fr)
**Abstract**
Coccolithophores are unicellular, calcifying marine algae that play an important role in the oceanic carbon
cycle via their cellular processes of photosynthesis (a $CO_2$ sink) and calcification (a $CO_2$ source). Alongside
the well-known, shallow-water coccolithophore blooms visible from satellites, deep niches of
coccolithophores are a poorly known but potentially important coccolithophore ecosystem. We
investigated the conditions that regulate the development of a deep coccolithophore niche (150-200 m
depth) along the BIOSOPE transect in the South Pacific oceanic gyre. We carried out batch culture
experiments with a coccolithophore strain isolated from the BIOSOPE transect, reproducing the in situ
conditions of light- and nutrient- (nitrate and phosphate) limitation. By simulating coccolithophore
physiology using an internal stores (Droop) physiological model, we were able to constrain fundamental
physiological parameters for this BIOSOPE coccolithophore strain. We show that simple batch experiments,
in conjunction with physiological modelling, can provide reliable estimates of fundamental physiological
parameters that are usually obtained in more time consuming and costly chemostat experiments. The
combination of culture experiments, physiological modelling and in situ data from the BIOSOPE cruise show
that coccolithophore growth in the deep BIOSOPE niche is co-limited by availability of light and nitrate. This
study contributes to the understanding of *Emiliania huxleyi* physiology, metabolism and behavior in a
disadvantageous ecosystem of the ocean.
**Keywords**
Coccolithophores, batch cultures, Deep niche, South Pacific Gyre, Droop model.



## 1. Introduction

Coccolithophores are unicellular photosynthetic and calcifying organisms that play a key role in the global carbon cycle (Paasche, 2001; Winter and Siesser, 1994). Through photosynthesis they participate to the upper ocean carbon pump ($CO_2$ sink), while via calcification they participate to the carbonate counter-pump ($CO_2$ source) (Paasche, 2001; Westbroek et al., 1993). The relative importance of calcification and photosynthesis dictates the effect coccolithophores have on ocean-atmosphere $CO_2$ fluxes (Shutler et al., 2013). Environmental conditions such as temperature, irradiance, nutrient concentrations and $pCO_2$ exert a primary control on the calcification/photosynthesis ratio (PIC:POC); and also affect coccolithophore biogeography via their influence on cellular growth rates. Together, these effects modulate the impact of coccolithophores on ocean-atmosphere $CO_2$ fluxes.

Despite the fact that certain coccolithophores have been fairly extensively studied in the laboratory (Daniels et al., 2014; Iglesias-Rodriguez et al., 2008; Krug et al., 2011; Langer et al., 2012; Rouco et al., 2013),  the factors controlling their biogeography in the global ocean are poorly understood (Boyd et al., 2010). In controlled laboratory conditions, coccolithophore growth is monitored as given environmental parameters are varied (Buitenhuis, 2008; Feng et al., 2008; Fritz, 1999; Langer et al., 2006; Leonardos and Geider, 2005; Paasche, 1999; Trimborn et al., 2007). In the ocean, geographical surveys of coccolithophore abundance and concomitant measurements of environmental variables contribute to defining coccolithophore biogeography in relation to the environment (Claustre et al., 2008; Henderiks et al., 2012). However, extrapolation of results from laboratory experiments to interpret field distributions is not straightforward, primarily because multiple environmental variables co-vary spatially in the ocean (Henderiks et al., 2012; Poulton et al., 2014).

In this respect, one of the least well understood, but possibly globally relevant niches where coccolithophores can be relatively abundant is that occurring at the deep nutricline of oceanic gyres. The deep niche of coccolithophores discovered during the BIOSOPE cruise in the South Pacific Gyre (Beaufort et al., 2007; Claustre et al., 2008) is probably the best studied example. This deep coccolithophore niche occurred at around the 200 m nutricline, at very low irradiance levels (< 20 $\mu mol.m^{-2}.s^{-1}$) with dissolved nitrate ($NO_3$) and phosphate ($PO_4$) concentrations of about 1 $\mu M$ and 0.2 $\mu M$, respectively, and was dominated by coccolithophore species belonging the family Noëlaerhabdaceae: *Emiliania huxleyi* , and several species of G*ephyrocapsa* and *Reticulofenestra* (Beaufort et al., 2007). Deep-dwelling (> 80 m depth) coccolithophores have also been observed in other geographic regions. Okada and McIntyre (1979) observed coccolithophores in the North Atlantic Ocean down to a depth of 100 m where *Florisphaera profunda* dominates coccolithophore assemblages in summer and *E. huxleyi* for the rest of the year. Coccolithophore assemblages dominated by *F.profunda* in the lower photic zone (LPZ > 100 m) of the subtropical gyres were observed by Cortés et al. (2001) in the Central North Pacific Gyre (station ALOHA)



and by Haidar and Thierstein (2001) in the Sargasso Sea (North Atlantic Ocean). Jordan and Winter (2000)
reported deep populations of coccolithophores dominated by *F. profunda* in the LPZ in the north-east
Caribbean but also reported a high abundance and co-dominance of *E. huxleyi* and *G.oceanica* through the
water column down to the top of the LPZ. These deep-dwelling coccolithophores are not recorded by
satellite-based remote sensing methods (Henderiks et al., 2012; Winter et al., 2014) that detect back-
scattered light from coccoliths from a layer only a few tens of meters thick at the surface of the ocean
(Holligan et al., 1993; Loisel et al., 2006).

Understanding the development of deep coccolithophore populations in low nutrient, low irradiance
environments would contribute to building a global picture of coccolithophore ecology and biogeography.
Laboratory cultures with coccolithophores that combine both nutrient and light limitation, however, are
scarce. One reason is that investigating phytoplankton growth under nutrient limitation in laboratory
experiments is complicated (Langer et al., 2013). In batch cultures the instantaneous growth rate decreases
as nutrients become limiting, making it hard to extract the dependence of growth rate on nutrient
concentrations (Langer et al., 2013). Chemostat cultures, where growth rates and nutrient concentrations
are kept constant under nutrient-limited conditions, offer an alternative (Engel et al., 2014; Leonardos and
Geider, 2005; Müller et al., 2012). Physiological parameters obtained in chemostat experiments have been
used in biogeochemical models to investigate environmental controls on phytoplankton biogeography
(Follows and Dutkiewicz, 2011; Gregg and Casey, 2007). Unfortunately, despite their relevance to nutrient
limited growth, chemostat cultures are more expensive, time-consuming and complicated to set up than
batch cultures (LaRoche et al., 2010).

In this paper, we investigate the growth of the coccolithophore *E. huxleyi* under light and nutrient co-
limitation with the aim of understanding the environmental controls on the development of deep
populations of this species discovered during the BIOSOPE cruise (Beaufort et al., 2007). We carried out
batch culture experiments with an *E. huxleyi* strain isolated during the BIOSOPE cruise and reproduced low
light and low nutrient conditions approaching the in situ values of the deep ecological niche. We monitored
the nitrogen and phosphorus content of particulate organic matter, as well as cell, coccosphere and
coccolith sizes, because these parameters are known to vary systematically with nutrient limitation (Fritz,
1999; Kaffes, 2010; Rouco et al., 2013). To overcome the conceptual limitations inherent in nutrient-limited
batch experiments (Langer et al., 2013), we modelled the transient growth conditions in the batch reactor
assuming that assimilation of nutrients and growth are either coupled (Monod, 1949) or decoupled (Droop,
1968) processes in the coccolithophore *E. huxleyi*. An independent check of our modelling approach was
obtained by also modelling the *E. huxleyi* batch culture data of Langer et al. (2013). Our joint culture and
modelling approach provides information on the conditions that control the growth of *E. huxleyi* in the
deep ecological niche of the South Pacific Gyre, and demonstrate that batch experiments, if conducted



thoroughly, may provide valuable estimates of fundamental physiological parameters that are otherwise
obtained via more time-consuming and costly chemostat experiments (LaRoche et al., 2010).

## 2. Materials and methods

### 2.1 Experimental

#### 2.1.1 Growth medium and culture conditions

Natural seawater collected near the Roscoff Biological Station (Brittany, France) was sterile filtered and
enhanced to K/2(-Si,-Tris) medium according to Keller et al. (1987). *Emiliania huxleyi* strain RCC911, isolated
in summer 2004 from a sample collected at 10 m depth near the Marquesas Islands during the BIOSOPE
cruise, was grown in batch cultures. Cells were acclimated to experimental conditions for at least three
growth cycles. Experiments were conducted in triplicate in 2.7 litre polycarbonate bottles (Nalgene) with no
head space, under a 12:12 light:dark irradiance cycle and at a temperature of 20°C. The concentration of
dissolved nitrate ($NO_3$) and phosphate ($PO_4$) and the irradiance levels were chosen to reproduce the
conditions prevalent in surface waters and at the nitricline of the oligotrophic gyre in the South Pacific
Ocean (Morel et al., 2007). The high light condition was 140 $\mu mol.m^{-2}.s^{-1}$ and the low light condition was 30
$\mu mol.m^{-2}.s^{-1}$. The low irradiance condition was at the upper end of the irradiance range (10-30 $\mu mol.m^{-2}.s^{-1}$)
in the deep BIOSOPE coccolithophore niche and we chose not to run experiments at irradiance levels lower
than this to avoid very long experimental runs. Nutrient concentrations at the beginning of batch
experiments were 100 $\mu M$ and 2.5-5.1 $\mu M$ for nitrate and 6.25 and 0.45-0.55 $\mu M$ for phosphate in nutrient-
replete and nutrient-limited conditions, respectively. For each irradiance level, three experiments were
carried out (in triplicate): control (nutrient-replete), phosphate limited (P-limited) and nitrate limited (N-
limited) conditions.

#### 2.1.2 Cell enumeration and growth rate

The growth of batch cultures was followed by conducting cell counts every day or every other day using a
BDFacs Canto II Flowcytometer. Experiments were stopped before the cell density reached ca. $1.5*10^5$
cells.mL$^{-1}$ in order to minimize shifts in the dissolved inorganic carbon (DIC) system. Cultures remained in
the exponential growth phase throughout the duration of the control (nutrient-replete) experiments. In
these control cultures, the growth rate ($\mu$) was obtained by conducting a linear regression of the cell
density data on the logarithmic scale. Nutrient-limited experiments were allowed to run until growth
stopped. The growth rate in nutrient limited conditions decreases in time as nutrients are depleted and it is
therefore not possible to calculate growth rate by means of regression analysis (Langer et al., 2013). The
dependence of growth rate on nutrient concentration in nutrient-limited conditions was investigated with
the numerical model introduced in Sect. 2.2 below.



### 2.1.3 Cell, coccosphere diameter and coccolith length

Samples were taken at the end of the experiments to acquire images of cells using an optical microscope
(x100, oil immersion, Olympus BX51 microscope). The internal cell diameter of 100 cells was measured for
each experimental culture using the ImageJ software (http://rsbweb.nih.gov/ij/). Images of coccospheres
and coccoliths were obtained with scanning electron microscopy (SEM). For SEM observations, samples
were filtered onto 1.2 μm polycarbonate filters (Millipore), rinsed with a basic solution (180 μL of ammomia
solution 25 % in a liter of MilliQ water) and dried at 55°C for 1 h. After mounting on an aluminum stub, they
were coated with gold-palladium and images were taken with a Phenom G2 pro desktop Scanning Electron
Microscope. For each experimental culture 100 coccospheres were measured using ImageJ. Three hundred
coccoliths per sample were measured using a script (Young et al., 2014) that is compatible with ImageJ in
order to measure the distal shield length (DSL) and coccolith width.

### 2.1.4 Dissolved inorganic carbon (DIC) and nutrient analyses

Subsamples for $pH_T$ (pH on the total scale), DIC and nutrient analyses were taken from culture media at the
beginning and at the end of each experiment. The pH was measured with a pHmeter-potentiometer
pHenomenal pH1000L with a Ross ultra combination pH electrode on the total scale and was calibrated
with a TRIS buffer. Samples for the determination of DIC were filtered through pre-combusted glass-fibre
filters (Whatman GF/F) into acid-washed glass bottles and poisoned with mercuric chloride. Bottles were
stored at 4°C prior to analysis. A LICOR7000 $CO_2/H_2O$ gas analyzer was used for the DIC analysis (precision ±
2 μmol.kg$^{-1}$). A culture aliquot (100 mL) was filtered onto pre-combusted glass-fibre filters (Whatman GF/F)
and then stored at -20°C in a polyethylene flask until nutrient analysis. Nitrate and phosphate
concentrations were measured using a CHN Auto analyzer Seal Analytical AAIII (detection limits were 0.003
μM for $PO_4$ and 0.01 μM for $NO_3$).

### 2.1.5 POC, PON, PIC, POP

For particulate organic carbon (POC), particulate organic nitrogen (PON), particulate inorganic carbon (PIC)
and particulate organic phosphorus (POP) analyses, samples (200 or 250mL) were filtered onto pre-
combusted (4 h at 450°C) glass-fibre filters (Whatman GF/F) and preserved at -20°C. POC and PON were
measured on the same filter that was dried overnight at 50°C after being placed in a fuming hydrochloric
acid dessicator for 2 h to remove coccolith calcite. POC and PON were analyzed using a NC Analyzer Flash
EA 1112. PIC was obtained using a 7500cx Agilent ICP-MS by analyzing the calcium concentration on the
glass-fibre filter (Whatman GF/F) extracted by a solution of hydrochloric acid. POP was determined as the
difference between the total particulate phosphorus and the particulate inorganic phosphorus, analyzed
according to the method of Labry et al. (2013).





### 2.2.1 Monod and Droop model
### *2.2 Modelling*
Growth of *E. huxleyi* in the batch reactors was simulated using Monod and Droop models of cellular growth.
In the Monod model (Monod, 1949), the growth rate depends on the external nutrient concentration and is
calculated as:
$$\mu = \mu_{\max} \cdot \frac{[N]}{[N] + K_N} \tag{1}$$


where $\mu_{max}$ (in day$^{-1}$) is the maximum growth rate in nutrient-replete conditions, $K_N$ (in $\mu$mol.L$^{-1}$) is the
(Monod) half-saturation constant for growth and *[N]* (in $\mu$mol.L$^{-1}$) is the concentration of nutrient N in the
batch reactor. Both $\mu_{max}$ and $K_N$ were obtained by fitting the model to the data, while *[N]* was the nutrient
concentration measured in the culture experiments.
Two differential equations keep track of the total cell abundance in the batch reactor (*Cells*) and the
limiting nutrient concentration in the reactor:
$$\frac{dCells}{dt} = \mu \cdot Cells \tag{2}$$


$$\frac{d[N]}{dt} = \frac{-N_{FIX} \cdot Cells}{V_R}$$

where $V_R$ (in litres) is the volume of the batch reactor, *Cells* (in cell.mL$^{-1}$) is the cell density measured during
the experiments, and $N_{FIX}$ the cell-specific N fixation rate (in $\mu$mol$_N$.cell$^{-1}$.day$^{-1}$) given by:
$$N_{FIX} = \mu \cdot Q_N \tag{3}$$


where $Q_N$, the (constant) cellular quota of nutrient N (in $\mu$mol$_N$.cell$^{-1}$), was calculated from the cellular
carbon quota, $Q_C$ (in $\mu$mol$_C$.cell$^{-1}$) (measured in experiments) and the C/N ratio obtained by the Redfield
ratio (Redfield, 1963).

In the Droop model (Droop, 1968) nutrient uptake and cellular growth are decoupled and cellular growth
depends on the internal store of the limiting nutrient. The time-dependent rate of nutrient uptake, $N_{up}$ (in
$\mu$mol$_N$.cell$^{-1}$.day$^{-1}$), is simulated using Michaelis-Menten uptake kinetics:
$$N_{up} = S_{cell} \cdot V_{\max} \cdot \frac{[N]}{[N] + K_N} \tag{4}$$

where $S_{Cell}$ (in $\mu$m$^3$) is the surface area of the cell, $V_{max}$ (in $\mu$mol$_N$.$\mu$m$^{-2}$.d$^{-1}$) is the maximum surface-
normalized nutrient uptake rate (obtained by fitting the model to the data) and $K_N$ (in $\mu$mol.L$^{-1}$) is the



(Michaelis-Menten) half-saturation constant for uptake of nutrient N. The volume and surface of cells ($S_{cell}$)
was either obtained by measurements of cells (both in the control culture and at the end of the nutrient-
limited cultures) or was estimated from $Q_C$, the cellular organic carbon quota (in $pmol_C.cell^{-1}$), and the
density of carbon in coccolithophore biomass (approximately equal to 0.015 $pmol_C$ $\mu m^{-3}$) (Aloisi, 2015).
The phytoplankton growth rate $\mu$ (in $d^{-1}$) was calculated based on the normalized $^n$Quota equation reported
in (Flynn, 2008):

$$\mu = \mu_{max} \cdot \frac{(1+KQ) \cdot (Q - Q_N^{min})}{(Q - Q_N^{min}) + KQ \cdot (Q_N^{max} - Q_N^{min})} \tag{5}$$

where $\mu_{max}$ (in $days^{-1}$) is the maximum growth rate attained at the maximum nutrient cell quota $Q_N^{min}$ (in
$\mu mol$ $cell^{-1}$), $Q_N^{min}$ (in $\mu mol$ $cell^{-1}$) is the minimum (subsistence) cellular quota of nutrient N below which
growth stops and $KQ$ is a dimensionless parameter that can be readily compared between nutrient types
and typically has different values for $NO_3$ and $PO_4$ (Flynn, 2008). While $Q_N^{max}$ and $Q_N^{min}$ were obtained from
the analyses of the particulate organic quota of the nutrient ($NO_3$ or $PO_4$) at the beginning and at the end of
the experiments, $KQ$ was obtained from fitting the model to the data. Thus, the growth rate depends on
the internal cellular quota of nutrient N, rather than on the external nutrient concentration like in the
Monod model of phytoplankton growth.
Three differential equations keep track of the total cell abundance in the batch reactor (*Cells*), the nutrient
concentration in the reactor (*[N]*, in $\mu mol.L^{-1}$) and the internal cellular quota of nutrient ($Q_N$, in $\mu mol.$ $cell^{-1}$):

$$\frac{dCells}{dt} = \mu \cdot Cells \tag{6}$$

$$\frac{d[N]}{dt} = \frac{-N_{up} \cdot Cells}{V_R} \tag{7}$$

$$\frac{dQ_N}{dt} = N_{up} - \mu \cdot Q_N \tag{8}$$

These three differential equations are integrated forward in time starting from initial conditions chosen
based on experimental values of the number of cells, nutrient concentration at the beginning of the
experiment and the cellular nutrient quota determined during growth in nutrient-replete conditions.
The dependence of the maximum growth rate on irradiance was determined independently by fitting the
growth rate determined in the exponential growth phase in our experiments and in the experiment of
Langer et al. (2013) to the following equation from MacIntyre et al. (2002):



$$\mu = \mu_{\max}\left(1 - e\left(\frac{-Irr}{KIrr}\right)\right)$$    (9)

where KIrr is the light-saturation parameter of growth in $\mu mol.m^{-2}.s^{-1}$ (MacIntyre et al., 2002; Fig. 1).

### 2.2.2    Modelling strategy
The Droop model presented here does not take into account the variation of size of coccolithophore cells
between the different experiments.
This model has eight parameters. Four are considered to be known and constant for a given experiment:
batch volume $V_R$, cell volume (and surface area $S_{Cell}$), and minimum and maximum cellular quota of
nutrient, respectively $Q_{min}$ and $Q_{max}$. The unknown parameters (the physiological parameters of interest)
are: the (Michaelis-Menten) half-saturation constant for nutrient uptake $K_{N/P}$, the maximum surface-
normalized nutrient uptake rate $V_{max}$, the maximum growth rate $\mu_{max}$ and the dimensionless parameter $KQ$
(N or P). The Monod model has fewer known parameters: batch volume $V_R$ and cellular quota of nutrient
$Q_{N/P}$. Unknown parameters are: maximum growth rate $\mu_{max}$ and the (Monod) half-saturation constant for
growth $K_{N/P}$.
The time-dependent cell density, limiting nutrient concentration and cellular particulate organic nitrogen
and phosphorus calculated by the models were fitted to the same quantities measured in the experiments.
For our experiments there were only two nutrient data points, one at the beginning and one at the end of
the experiments. We artificially inserted a third nutrient-quota data point at the end of the exponential
growth phase, setting it equal to the nutrient quota at the beginning of the experiment. In this way the
model is forced to keep the nutrient quota unchanged during the exponential growth phase. This is a
reasonable assumption, as cellular nutrient quotas should start to be affected only when nutrient
conditions become limiting.
The quality of the model fit to the experimental data was evaluated with a cost function. For a given model
run, the total cost function was calculated as follows:
$$TotCost = \sum_{i=1}^{n} (\Delta x_i)^2$$    (10)
where n is the number of data points available and $\Delta x_i$ is the difference between the data and the model
for the $i^{th}$ data point:
$$\Delta x_i = Data(x_i) - Model(x_i)$$    (11)
where $x_i$ is the data or model value for the considered variable (cell density, limiting nutrient concentration
or cellular limiting nutrient quota). The lower the cost function is, the better the quality of the model fit to



the data. For a given experiment, the best-fit of the model to the data was obtained by running the model
repeatedly imposing a high number of combinations of input parameters (typically 500000 model runs for
every experiment) and selecting the parameter setting that yielded the lowest cost.

## 3. Results

### 3.1  Laboratory experiments with E. huxleyi strain RCC911

#### 3.1.1    Cell density and growth rate

Growth curves for all experiments with *E. huxleyi* strain RCC911 are shown in Fig. 2. Experiments run in
high light conditions attained target cell densities (in nutrient-replete, control experiments) or nutrient
limitation (in nutrient-limited experiments) in a shorter time compared to experiments run in low light
conditions. Growth in nutrient-replete cultures in both light conditions followed an exponential growth
curve (growth rates in the control nutrient-replete experiments were $0.91 \pm 0.03$ $d^{-1}$ and $0.28 \pm 0.01$ $d^{-1}$ for
the high light and low light experiments, respectively; Table 1) whereas in nutrient-limited experiments
growth evolved from an exponential to a stationary phase at the end of the experiment, except the P-
limited culture at low light where the stationary phase was not attained (growth rate of $0.13 \pm 0.01$ $d^{-1}$).

#### 3.1.2    Dissolved nutrients, $pH_T$ and DIC

In the high light experiment, $NO_3$ concentration decreased to $0.18 \pm 0.03$ µM in N-limited cultures and $PO_4$
concentration decreased to $0.011 \pm 0.004$ µM in P-limited cultures at the end of the experiments, and in
low light conditions the final $NO_3$ and $PO_4$ concentrations were $0.13 \pm 0.02$ µM and $0.008 \pm 0.006$ µM,
respectively (Table 1). Thus, nutrients where nearly completely exhausted at the end of our nutrient-limited
experiments. Seawater carbonate chemistry was quasi-constant over the course of the experiments in all
treatments, with as reported by Langer et al. (2013), the P-limited cultures showing the biggest change in
DIC (12-13%; Table 1).

#### 3.1.3    Carbon, nitrogen and phosphorus cell quotas and ratios

Compared to the control experiments, cellular POC, PIC and PON quotas increased in the P-limited cultures
at both light levels,, while cellular POP quota decreased (Table 2; Fig. 3). In the N-limited cultures, cellular
PIC and POC quotas increased, with the exception of POC at low light that remained nearly unchanged,
while cellular PON and POP quotas decreased at both light levels. N-limiting conditions resulted in a
decrease of the PON:POC ratio in both light regimes (Fig. 4A, Table 2). Changes in the POP:POC ratio (Fig.
4*Figure 4*B, Table 2) were harder to discern due to a large error bar in high light and nutrient-replete
conditions. Notwithstanding, POP:POC was lower in P-limited experiments compared to nutrient-replete
experiments. The PIC:POC ratio increased with both N- and P-limitation (Fig. 4C) at both light regimes. For
the high light experiment, the PIC:POC ratio was highest in the P-limited culture ($0.52 \pm 0.14$), while in the





low light conditions, the highest ratio was recorded in the N-limited culture (0.33 ± 0.02) (Fig. 4C).
Light limitation led almost invariably to a decrease in POC and PIC, with the exception of POC in nutrient-
replete conditions, for which the decrease in irradiance did not induce a decrease (Table 2, Fig. 3). In P-
limited cultures POP and PON decreased with light limitation, whereas in N-limited cultures POP and PON
increased with light limitation (Fig. 3). With the exception of the POP:POC ratio in P-limiting conditions that
was not affected by the change in light regime, both PON:POC and POP:POC ratios increased with light
limitation. In all three nutrient conditions, the PIC:POC ratio decreased with light limitation.

### 306    3.1.4    Cell, coccosphere and coccolith size

Cell size varied with both nutrient and light limitation (Table 3). In high light conditions the cell volume was
higher for the P-limited culture, with a volume of 77.2 ± 19.9 $\mu m^3$, than for the control culture and the N-
limited culture that had similar cell volumes (ca. 47-50 $\mu m^3$). Cell volume was consistently lower in low light
conditions, and P-limited cultures again had higher cell volume than in the control and N-limited cultures.
POC content and cell volume is illustrated in Fig. 5A ($R^2$=0.75, p<0.006, n=8).
P-limitation resulted in higher coccosphere volume and higher DSL than the other nutrient conditions in
both light regimes (Table 3). For example, the coccosphere volume in high light was 260 ± 88 $\mu m^3$ for the P-
limited experiment, whereas it was 109 ± 23 $\mu m^3$ for the control experiment and 139 ± 41 $\mu m^3$ for the N-
limited experiment. There was no measurement of coccosphere volume and DSL in the low light control
culture because of a lack of cells on the filters. However, the coccosphere volume for P-limited in low light
conditions followed the same trend as the cell size, a decrease with lower light. Figure 5B shows the
correlation between cell and coccosphere volume ($R^2$=0.90, p<0.002, n=7). The correlations between DSL
and coccosphere size ($R^2$=0.73, p<0.05, n=7) and between the DSL and the cell size ($R^2$=0.85, p<0.003, n=7)
are illustrated in Fig. 6.
The thickness of the coccolith layer, calculated by subtracting the cell diameter from the coccosphere
diameter and dividing by two, was higher for P-limited cultures in both light conditions: 1.294 ± 0.099 $\mu m$
for high light and 1.02 ± 0.043 $\mu m$ for low light compared with the other cultures which were between 0.66
and 1 $\mu m$. This observation is consistent with the high PIC quota and relatively large size of coccospheres
and coccoliths of *E. huxleyi* under P-limitation.

### 327    *3.2 Modelling results*


We applied the modelling approach to both the data from our batch culture experiments with strain
RCC911 and to the batch culture data of Langer et al. (2013) who tested N- and P-limited growth of *E.*
*huxleyi* strain PML B92/11 cultured in high light conditions (400 $\mu mol.m^{-2}.s^{-1}$), optimal temperature (15°C)
and quasi-constant carbon system conditions. Measurements of cell density, nutrient concentrations and
cellular particulate matter from both sets of experiments were used for the present modelling study.





The measured minimum PON value (5.71 fmol cell$^{-1}$) for the N-limited experiment of Langer et al. (2013) is
very low compared with the PON quota in other N-limited *E. huxleyi* experiments reported in the literature
(38.9-39.3 fmol.cell$^{-1}$ in Sciandra et al. (2003) and 51.4 fmol.cell$^{-1}$ in Rouco et al. (2013)). When the $Q_N^{min}$
value of Langer et al. (2013) was used in the model, the model fit to the experimental data degraded
considerably (data not shown). Consequently, we decided to recalculate $Q_N^{min}$ using the initial
concentration of dissolved N and the final cell density in the reactor (column "Calculation"in Table 4). This
calculated value of $Q_N^{min}$, that in all cases except for the N-limited experiments of Langer et al. (2013) was
very similar to the measured minimum PON quota, was comparable to values reported in the literature for
*E. huxleyi* and resulted in a very good fit of the model to the experimental data. To be coherent, we applied
this approach to all values of $Q_N^{min}$ and $Q_P^{min}$ used in the modelling exercise.

The Droop model was able to accurately reproduce both experimental data sets (Fig. 7 to 11), whereas the
Monod model was not able to reproduce the rise in cell number after the limiting nutrient had been
exhausted (Fig. 7). The modelling approach allows evaluation of the evolution of experimental variables
that are complicated to determine analytically (Fig. 7 to 11), i.e.: 1) the nutrient-uptake rate, that follows
the same trend as the nutrient concentration in the reactor; (2) the limited-nutrient/C ratio, that  starts at a
maximum value, stays constant during the duration of the exponential phase and then declines due to the
exhaustion of external nutrient, reaching a minimum as the culture attains the stationary phase, and (3) the
instantaneous growth rate, that follows the trend of the limiting nutrient ratio, reaching zero when the
culture attains the stationary phase.

The values for the best-fit for physiological parameters obtained by applying the Droop model to our
experiments with *E. huxleyi* strain RCC911 and to the experiments of Langer et al. (2013) are presented in
Table 4. Overall, the best-fit values for the two strains were similar, suggesting that the modelling approach
is sound. Values for the Monod nutrient assimilation constant $K_N$ determined in our experiments and in
those of Langer et al. (2013) were comparable. However, for $K_P$, the value was consistent between our high
and low light experiments, but considerably lower for the Langer et al. (2013) experiment. The same holds
true for the maximum surface nutrient-uptake rate $V_{max}$ (except for our P-limited low light experiment). The
dimensionless parameters $KQ_N$ and $KQ_P$ were comparable between the two studies (for high light
conditions) and in both cases $KQ_P$ was higher than $KQ_N$. Maximum growth rates in high light conditions
were similar for both N-limited and P-limited experiments. As expected, the maximum growth rate for our
low light cultures was considerably lower.
To test the reliability of the model to obtain estimates of the physiological parameters, we forced the
model to run with a range of values for a given parameter, while letting the other three parameters vary
over a wide range, obtaining plots of the value of the cost function (Eq. 9) as a function of the value of the
imposed parameter. The process was repeated separately for the four unknown parameters (Fig. 12) shows



the results for the N-limited culture of Langer et al. (2013). For all of the parameters except for $K_{N/P}$, this
exercise yielded a U-shaped curve with a minimum of the cost function corresponding to the best-fit
parameter values presented in Table 4. This shows that the model is well suited to find a best-fit value of
these parameters. Three minima of the cost function were found for $K_{N/P}$ (Fig. 12) of which only the lowest
was consistent with values reported in the literature (Riegman et al., 2000). This value was chosen to obtain
the best-fit of the model to the experimental data.
**4. Discussion**
*4.1 Batch culture experiments*

The batch culture experiments presented here provide new insights into the physiology of the ecologically
dominant coccolithophore *E. huxleyi* under conditions of light and nutrient limitation that are relevant for
the study of deep coccolithophore niches. Leonardos and Geider (2005) carried out cultures in low light and
low phosphate conditions, but they did not measure PIC and thus did not report PIC:POC ratios. The culture
study reported here is thus the first experiment where changes in the PIC:POC ratio due to light-limitation
are explored for nutrient-limited cultures.
In our experiments, cultures were harvested at relatively low cell densities (maximum of ca. $1.6*10^5$
cells.mL$^{-1}$ in the P-limited low light experiment, and $< 1.3*10^5$ cells.mL$^{-1}$ in all other treatments) in order to
ensure that changes in the carbonate system were within a minimal range (generally < 10%; Table 1) that is
not expected to have a significant influence on measured physiological parameters (Langer et al., 2007;
LaRoche et al., 2010). Hence, it can be stated that the observed phenomena stem from N-/P-limitation
and/or light limitation (depending on the treatment) rather than from carbon limitation.
Comparison of the growth curves illustrated in Fig. 2 demonstrates that growth limitation was attained in
both our low nutrient and low light treatments relative to control (high nutrient / high light) conditions.
Consistent with previous experimental results (Langer et al., 2013; Leonardos and Geider, 2005; Müller et
al., 2012; Oviedo et al., 2014; Rouco et al., 2013) the relatively low cellular PON or POP quotas (and
PON:POC and POP:POC ratios) at the end of the low nutrient experiments indicate that nutrient limitation
of growth occurred in our low nutrient experiments. The stationary phase was not attained in the
phosphate-limited low light culture, but the very low POP quota (and POP: POC ratio) and increased cell
size indicate that P-limitation was starting to significantly affect cellular physiology, showed as well by the
decrease in growth rate between the nutrient-replete conditions in low light and this experiment.

In nutrient-replete conditions, low light had no effect on POC quota (Fig. 3) and cell size (Fig. 5) within the
limit of uncertainty of measurements. In same nutrient condition, low light did however cause a decrease in
PIC quota (and therefore a decrease in the PIC:POC ratio). Although the same observation will be done for
nutrient-limited conditions in low light in the following paragraphs, the quota for nutrient-replete



conditions in low light is unexpectedly small and the lack of measurements for coccospheres and coccoliths
size show a potential problem in the calcification process for this experiment.

In high light, N-limited cells were roughly the same size as nutrient-replete cells but had higher POC quota.
However, large error margins do not allow explaining these observations as the density in pg.µm$^{-3}$
calculated shows no significant difference between experiments. In low light, N-limited cells were similar in
size and POC quota to nutrient-replete cells. According to Müller et al. (2008), N-limited cells decrease in
volume due to substrate limitation and lower assimilation of nitrogen in the G1 phase of the cell division
cycle, but in our experiments N-limitation did not cause a decrease in cell volume. N-limitation led to an
increase in PIC quota relative to nutrient-replete cells in both high and low light conditions (Fig. 3) (Table 3).
Fritz (1999) reported as well an increase of PIC content in N-limited conditions as Müller et al. (2008).
However, Raven and Crawfurd (2012) affirm that N-limitation lead to a reduced biomass rather than calcite
which lead as well to a decreasing in cell size. Coccoliths in N-limited cultures tended to be smaller (in line
with the observations of Paasche, 1998) and coccosphere volumes tended to be higher than in nutrient-
replete cultures (large uncertainties of measurements), so N-limited cells presumably produced more
coccoliths. N-limitation led to an increase in the PIC:POC ratio in both high and low light conditions, a result
that is consistent with most previous N-limitation studies with *E. huxleyi* (see review by Raven and
Crawfurd, 2012).

P-limitation had the greatest effect on cell size, cells being significantly larger under P-limitation than
control conditions in both high and low light conditions. The increase in cell volume was accompanied by
increases in both POC and PIC quotas (Fig. 3). According to Müller et al. (2008), P-limitation inhibits DNA
replication while biomass continues to build up, leading to an increase in cell volume. This could explain
why the volume of P-limited cells in high light was very high in our experiments, but our results indicate
that when growth was already limited by light availability, P-limited cells were not as large, had
correspondingly lower POC and POP quotas, and were able to continue dividing beyond the cell density at
which growth was limited in high light conditions, thus representing an interesting case of the combined
effects of co-limitation. P-limitation resulted in considerably higher coccosphere volume than the other
nutrient conditions, in line with the observations of Müller et al. (2008) and Oviedo et al. (2014). In high
light the PIC quota in P-limited cells was more than tripled relative to nutrient-replete conditions (general
effect of a phosphate limitation reported by Raven and Crawfurd (2012) (Table 2), likely due to the
occurrence of larger (as shown by high DSL values) and potentially more numerous coccoliths (Gibbs et al.,

437 2013).


In the P-limited experiment, PIC:POC ratios increased relative to nutrient-replete cultures, like in the
experiments of van Bleijswijk et al. (1994) and Berry et al. (2002), although Oviedo et al. (2014) reported



that the response of the PIC:POC ratio to P-limitation is strain-specific in *E. huxleyi*. The increase in PIC:POC
in *E. huxleyi* is often greater for decreasing phosphate than for decreasing nitrate (Zondervan, 2007), as was
the case in our high light experiment, but in low light the PIC:POC ratio was higher under N-limitation, again
highlighting that co-limitation can have unexpected physiological consequences.

The PIC:POC ratio decreased with light limitation in nutrient replete and nutrient limited conditions in our
experiment (Fig. 4). In a review of environmental effects on coccolithophore calcification, Zondervan (2007)
stated that due to the lower saturation irradiance for calcification than photosynthesis in *E. huxleyi*, the
ratio of calcification to photosynthetic C fixation increases with decreasing light intensities. However, due
to a more rapid decline of calcification relative to photosynthesis this ratio decreases again under strongly
light-limiting conditions (below approximately 30 $\mu mol.m^{-2}.s^{-1}$). This phenomena was also reported by
Paasche (1999) and Zondervan et al. (2002). Several other studies reported as well similar variations with *E.*
*huxleyi* but different strains: Rokitta and Rost (2012) found a decrease of PIC:POC ratio between 50 and 300
$\mu mol.m^{-2}.s^{-1}$ for a comparable $pCO_2$ as our experiments; same observations for Feng et al. (2008) who
reported also a decreasing ratio between 50 and 400 $\mu mol.m^{-2}.s^{-1}$; the light intensities experiments of
Trimborn et al. (2007) showed a decrease of the ratio with a decreasing light from 300 to 30 $\mu mol.m^{-2}.s^{-1}$;
and Rost et al. (2002) found an increase of the PIC:POC ratio between 15, 30 and 80 $\mu mol.m^{-2}.s^{-1}$, again for a
comparable $pCO_2$ as our experiments, and then a decrease of the ratio from 80 to 150 $\mu mol.m^{-2}.s^{-1}$. Our
results indicate that calcification was already more severely limited than photosynthesis at 30 $\mu mol.m^{-2}.s^{-1}$
in the strain RCC911.
The correlations between DSL and coccosphere size ($R^2$=0.73, p<0.05, n=7) and between DSL and cell size
($R^2$=0.85, p<0.003, n=7) are illustrated in Fig. 6 and the results are consistent with the correlation reported
by Gibbs et al. (2013) between coccoliths and coccospheres in fossil sediment samples. These observations
suggest that coccosphere and coccolith size in the water column and in sediments could be used as a proxy
for cell size and PIC quota (Aloisi, 2015).

**4.1.1    Summary effect of nutrient limitation**
Apart for the phosphate limited and low light experiment, nutrient limitation slowed down the cell division
and so brought the cell into a stationary phase at the end of the experiment. Nutrient limitation as well
decreased the particulate organic matter quota corresponding to the limited nutrient (POP for P-limitation
and PON for N-limitation) and increased the PIC:POC ratio for both light conditions. The effect of nutrient
limitation on morphological properties is more complicated to describe due to large error bar. However, an
increase in cell/coccosphere size was observed under P-limitation and almost no effect was reported by the
N-limitation.



### 4.1.2   Summary effect of light limitation

Light limitation decreased the PIC quota and the cell size tended to decrease (no measurement for
coccospheres and coccoliths size). PIC:POC ratio decreased with light limitation in every nutrient conditions
whereas PON:POC and POP:POC increased with light limitation. As reported by Zondervan (2007), further
investigations need to be carried out to improve the understanding of the effect of light intensity on the
PIC:POC ratio.


### 4.1.3   Summary effect of co-limitation

P-limitation induces an important increase in cell size which is less important in low light where cells were
able to continue to divide in contrast to the P-limited and high light experiment. These observations
support that the cells growth is less affected by the P-limitation in low light than in high light conditions.
PIC:POC was higher in N-limited and low light conditions whereas in high light conditions P-limited
experiment got the higher ratio.


### *4.2  E. huxleyi physiological parameters obtained by modelling growth in a batch reactor*


The Droop model (Fig. 6 to 8) was able to accurately reproduce the experimental data obtained using *E.*
*huxleyi* strain RCC911 as well as the data of Langer et al. (2013). The Monod model, however, was not able
to reproduce the rise in cell number after the limiting nutrient had been exhausted (Fig. 7). This shows that,
as for several other phytoplankton groups (Lomas and Glibert, 2000), *E. huxleyi* has the ability to store
nutrients internally to continue growth to some extent when external nutrient levels become very low. In
our experiments and those of Langer et al. (2013), cells grew on their internal nutrient reserves and
managed two to three cell divisions in the absence of external nutrients.


Numerous studies have estimated the maximum nutrient uptake rate $V_{max}$ and the half-saturation constant
for nutrient uptake $K_{N/P}$, especially for nitrate uptake, for a variety of phytoplankton species. The values
obtained in our study for $K_N$ for high light *E. huxleyi* cultures (Table 4) are comparable to those reported in
the literature. Using *E. huxleyi* in chemostat experiments, Riegman et al., (2000) found $K_N$ values between
0.18 and 0.24 μM and $K_P$ between 0.10 and 0.47 μM. In addition, they reported a $V_{maxN}$ of $7.4.10^{-6}$ μmol.cell$^{-1}$.d$^{-1}$
which is between the $V_{maxN}$ found for PML B92/11 and for RCC911 (Table 4).
When comparing physiological parameters between phytoplankton taxa, the scaling of physiological
parameters with cell size has to be taken into account (Marañón et al., 2013).
Marañón et al. (2013) plotted $Q_{min}$ and $\mu_{max}$ against cell size (see Fig. 13A for $Q_{min}$ versus cell size) for
different phytoplankton species. In these plots coccolithophores fall with the smallest diatoms. Figure 13B
reports $V_{maxN}$ versus cell size for different groups of phytoplankton based on the results of Litchman et al.
(2007) (using a compiled database) and of Marañón et al. (2013) (22 species cultivated) and the results



obtained with the Droop model in this study. Despite the different procedures used to obtain $V_{maxN}$
(simulated with a model or measured experimentally), all values for coccolithophores fall in the same
range.
Litchman et al. (2007) found a linear correlation between the maximum uptake rate and the half-saturation
constant for nitrate uptake as in Collos et al. (2005) across phytoplankton groups (Fig. 14). This correlation
defines a physiological trade-off between the capacity to assimilate nutrients efficiently (high $V_{max}$) and thus
grow rapidly, and the capacity to assimilate nutrients in low-nutrient environments (low $K_{N/P}$) and thus
thrive in oligotrophic conditions. This analysis shows that large phytoplankton like diatoms and
dinoflagellates have high maximum nitrate uptake rates and high half-saturation constant for nitrate
uptake. As reported by Cavender-Bares et al. (2001), small cells are mainly found in low nutrient
concentration environment whereas larger cells are more abundant in high nutrient environment.
*E. huxleyi* maximum uptake rate and half-saturation constant for nitrate uptake were found to be low
compared to the other groups and their maximum growth rate is amongst the highest which means that it
will be more abundant in low nitrate waters compared to diatoms and dinoflagellates (Litchman et al.,
2007). According to Gregg and Casey (2007), there is a high affinity of coccolithophores for low nutrient and
low light environments, whereas in high nutrient and high light environments, coccolithophores will be
disadvantaged compared to diatoms and cyanobacteria because of their high growth rate and higher
sinking rate, respectively. The ideal conditions for an optimal abundance of coccolithophores will be low
nutrient and low light areas with a parallel inhibition of the growth of diatoms and chlorophytes, but with
vertical mixing strong enough to avoid the sinking of cells (Gregg and Casey, 2007).

The Droop model presented in this paper provides a simple procedure to obtain fundamental physiological
parameters from batch culture experiments. Usually physiological parameters are obtained experimentally
using continuous laboratory cultures (chemostats) (Eppley and Renger, 1974; Terry, 1982; Riegman et al.,
2000; Müller et al., 2012). However, batch cultures are easier to carry out and require minimal equipment
and are hence often used for culture experiments (LaRoche et al., 2010).

### *4.3  Controls on E. huxleyi growth in the deep BIOSOPE niche*

The BIOSOPE cruise was carried out in 2004 along a transect across the South Pacific Gyre from the
Marquesas Islands to the Peru-Chili upwelling zone. The aim of this expedition was to study the biological,
biogeochemical and bio-optical properties (Claustre et al., 2008) of the most oligotrophic zone of the
world's ocean (Claustre and Maritorena, 2003). The deep ecological niche of coccolithophores along this
transect occurred at the Deep Chlorophyll Maximum (DCM) (Beaufort et al., 2007). According to Claustre et
al. (2008) and Raimbault et al. (2007), the nitrate concentration at the GYR station at the deep
coccolithophore niche (between 150 and 200m depth) was between 0.01 and 1 µM. In our nitrate-limited



low light culture experiment (Fig. 15), this concentration occurred between the end of the exponential
growth phase and the beginning of the stationary phase (days 8 to 9), when nitrate-limitation began to
affect instantaneous growth rates. Our phosphate-limited experiment did not proceed long enough for the
instantaneous growth rate to decrease appreciably, but Claustre et al. (2008) reported a nitrate
concentration <3 nM in the 0-100 m water column and a detectable phosphate concentration always above
0.1 μM in surface layers (Raimbault and Garcia, 2008) and Moutin et al. (2008) concluded that phosphate is
apparently not the limiting nutrient for phytoplankton along the BIOSOPE transect. The picture that
emerges is consistent with the model of Klausmeier and Litchman (2001), who predicted that limitation of
growth in a DCM should be limited by both light and one nutrient, with the upper layer of the DCM being
limited by nutrient supply and the deeper layer by light. However, the vertical diffusivity of nitrate through
the nitracline needs to take into account and could potentially bring dissolved nitrate into the deep niche of
coccolithophores (Holligan et al., 1984). The experiments and modelling work presented here allow us to
conclude that the growth of *E. huxleyi* in the deep ecological niche at the GYR station of the BIOSOPE
transect is clearly limited by the light and potentially limited by the dissolved nitrate, with *E. huxleyi* growth
in the upper part of the niche mostly co-limited by irradiance intensity and nitrate availability, whereas
irradiance is the main limiting factor in the lower part of the niche where nitrate becomes more
concentrated.
The depth-distribution of the modelled *E. huxleyi* growth rate, and of dissolved nitrogen, light intensity,
chlorophyll a concentration and coccolithophore abundance supports the inferred light-nitrate co-
limitation (Fig. 16). We used the physiological parameters constrained in our experiments together with a
steady state assumption for uptake and assimilation of nitrate (see appendix) to obtain the vertical profile
of *E. huxleyi* growth rate at the GYR station (Fig. 16). This calculation, forced by the irradiance and nitrate
data at the GYR station, shows that *E. huxleyi* growth rate is maximal at the depth of the maximum
chlorophyll a concentration. The half-saturation constant for nitrate uptake $K_N$ constrained with the Droop
model (0.09 μmol.L$^{-1}$) lies within the deep niche (Fig. 16). The maximum growth rate at the GYR station
(0.024 d$^{-1}$ at 175 m depth) corresponds to an *E. huxleyi* generation time of 29.28 days, suggesting that
division rate at the DCM is very slow (so slow that it would be difficult to reproduce in culture experiments).
This point highlights the importance of this growth rate calculation which provides a useful way to
investigate the growth potential of *E. huxleyi* in the DCM of the South Pacific Gyre. Moreover, as the growth
rate for the low light culture and replete conditions experiments was at 0.28 d$^{-1}$ with an irradiance of
approximately 30 μmol.m$^{-2}$.s$^{-1}$, it is not surprising that the potential growth rate in the deep niche was so
small at irradiance below 20 μmol.m$^{-2}$.s$^{-1}$ and low nitrate concentration.
With the above limitation pattern in mind, it is possible to predict the effect of nitrate and light variability
on the vertical changes in the *E. huxleyi* PIC:POC ratio in gyre conditions. According to our experimental
results, the PIC:POC ratio increases slightly with nitrate limitation but the strongest effect on PIC:POC ratio
seems to be supported by light intensity. As explain in the Sect. 4.1 of this discussion, several studies have



shown that the ratio increases with light decreasing above a range of light between 30 and 80 $\mu mol.m^{-2}.s^{-1}$
but decreases with a light limitation below this range. Considering Fig. 16, the PIC:POC ratio in *E. huxleyi*
would be expected to be  intermediate in surface waters (nitrate-poor but strongly high light intensity;
approximately 1800 $\mu mol.m^{-2}.s^{-1}$), then increases to attain a maximum value between the lower part of
subsurface waters and the beginning of the deep niche (between 80 $\mu mol.m^{-2}.s^{-1}$ and 30 $\mu mol.m^{-2}.s^{-1}$ ;
therefore between 110 m and 150 m depth), decrease in the lower part of the deep niche, and finally
decreases drastically  in deeper water, nitrate-rich but low-irradiance waters. This prediction cannot be
verified with the available published data from the BIOSOPE transect, but same conclusions for the upper
part of the ocean have been observed through in situ measurements by Fernández et al. (1993) and these
predictions could easily be verified in future expeditions to coccolithophore-bearing DCM zones.
Klaas and Archer (2002) reported that calcium carbonate is mainly exported by coccolithophores to the
deep sea and that the rain of organic carbon is mostly conducted by calcium carbonate because of its
higher density than opal and higher abundance than terrigenous material. Then a decrease of PIC quota by
low irradiance will decrease the calcium carbonate rain to the sediments related to *E. huxleyi*. However, the
cellular PIC quota is maybe decreasing but the PIC total increase in the deep niche compared to the upper
and deeper water column. In this context the effect on rain ratio, therefore on carbon pump and carbonate
counter-pump, needs to be integrated over the whole photic zone and considering the whole particulate
organic and inorganic matter.
**5.  Concluding remarks**

We present one of the few laboratory cultures experiments investigating the growth and PIC:POC ratio of
the coccolithophore *E. huxleyi* in light- and nutrient-limited conditions, mimicking those of the deep
ecological niche of coccolithophores of the South Pacific Gyre (Beaufort et al., 2007; Claustre et al., 2008).
By combining batch culture experiments with a simple numerical model based on the internal stores
(Droop) concept, we show that: 1) *E. huxleyi* has the capacity to divide up to three times in the absence of
external nutrients by using internal nutrient stores and is more affected by phosphate limitation in high
light than in low light conditions; 2) a simple batch culture experimental set-up, as opposed to the more
time-consuming and expensive continuous culture approach, can be used to obtain fundamental
physiological parameters, such as the maximum surface- normalized nutrient uptake rate and the half-
saturation constant for nutrient uptake, that describe the response of phytoplankton growth to
environmental conditions; 3) the position of the deep coccolithophore niche of the South Pacific Gyre was
defined by the maximum of coccosphere reported by Beaufort et al. (2007) and the limitation of growth in
this niche is the result of contrasting gradients of light (decreasing downwards) and nitrate (decreasing
upwards), studied through a combination of experimental results, Droop modelling and in situ data; and
confirming the theoretical prediction of Klausmeier and Litchman (2001).





**Appendix**

To obtain the growth rate through the vertical profile at the station GYR, we needed to express the cellular
quota $Q_N$ of a function of the nitrate concentration $NO_3$ *[N]*. To deal with this purpose, we resolve the
system of three equations from the Droop theory:

$$\frac{dQ_N}{dt} = N_{up} - \mu \cdot Q_N \tag{A1}$$

$$N_{up} = S_{cell} \cdot V_{max} \cdot \frac{[N]}{[N] + K_N} \tag{A2}$$

$$\mu = \mu_{max} \cdot \frac{(1 + KQ) \cdot (Q - Q_N^{min})}{(Q - Q_N^{min}) + KQ \cdot (Q_N^{max} - Q_N^{min})} \tag{A3}$$

Considering a stationary state (uptake-assimilation steady state) and thus assuming the differential Eq. (A1)
equal to zero, we resolve the system to express the cellular quota $Q_N$ versus the nitrate concentration (see
Fig. A1):
$$A = \frac{1}{2 \cdot (1 + KQ) \cdot \mu_{max} \cdot (K_N + [N])} \cdot \left( \left( K_N \cdot (1 + KQ) \cdot \mu_{max} \cdot Q_N^{min} \right) \right) \tag{A4}$$

$$B = \left( (1 + KQ) \cdot \mu_{max} \cdot [N] \cdot Q_N^{min} \right) + \left( [N] \cdot S_{cell} \cdot V_{max} \right) \tag{A5}$$

$$C = \sqrt{ \begin{array}{l} 4(1 + KQ) \cdot \mu_{max} \cdot [N] \cdot (K_N + [N]) \cdot \left( KQ \cdot Q_N^{max} - (1 + KQ) \cdot Q_N^{min} \right) \cdot S_{cell} \cdot V_{max} \\ + \left( (1 + KQ) \cdot \mu_{max} \cdot (K_N + [N]) \cdot Q_N^{min} + [N] \cdot S_{cell} \cdot V_{max} \right)^2 \end{array} } \tag{A6}$$

$$Q_N = A + B + C \tag{A7}$$

Thus, the growth rate can be express depending of the irradiance (and *KIrr*; see Sect. 2.2.1) and the cellular
quota $Q_N$. The other parameters are known (output of the model for the experiment reproducing the
condition of the nitracline):

$$\mu = \mu_{max} \cdot \frac{(1 + KQ) \cdot (Q - Q_N^{min})}{(Q - Q_N^{min}) + KQ \cdot (Q_N^{max} - Q_N^{min})} \cdot \frac{Irr}{Irr + KIrr} \tag{A8}$$



The vertical profile of the growth rate of coccolithophores at the station GYR, calculated with the previous
equation, is shown in the Fig. 16.

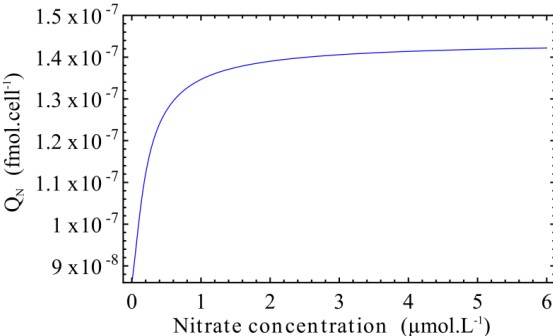


*Figure A1.* Cellular quota of nitrogen versus the nitrate concentration using parameters of the best-fit

results of the model ran for the low light and nitrate limited experiment with RCC911.



## Acknowledgements


This project was supported by the TELLUS CLIMAHUX project (INSU-CNRS) and the MODIF project of the
Institut Pierre Simon Laplace (IPSL). We thanks C. Schmechtig for having provide access to the BIOSOPE
data base, F. Le cornec and I. Djouraev for helping with the PIC analysis at the Institut de Recherche pour le
Développement (IRD) ALYSE platform and C. Labry and A. Youenou for carrying out the POP analysis at
IFREMER. We are grateful to C. Leroux for the analysis of our POC, PON samples and the research team
CHIM from the Station Biologique of Roscoff for their help: T. Cariou for the dissolved nutrients analysis and
de-carbonatation of the POC, PON samples; M. Vernet for help processing DIC samples; and Y. Bozec for
DIC analysis. We thank as well A. Charantonis for his statistical advices for the modelling part. The lead
author is supported by a doctoral fellowship from the French Minister of Education and Research (MESR).



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

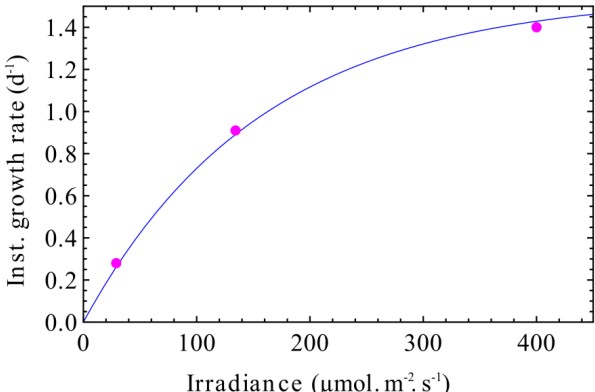

*Figure 1*. Instantaneous growth rate versus irradiance calculated with equation 9 (*KIrr* = 157 µmol.m$^{-2}$.s$^{-1}$).
Pink points represent the irradiance and growth rate of the experiments carried out by Langer et al. (2013)
and our experiments. The blue line is the result of equation 9.

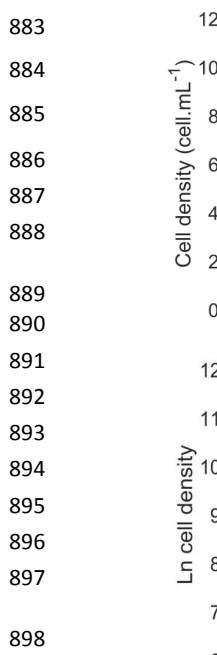
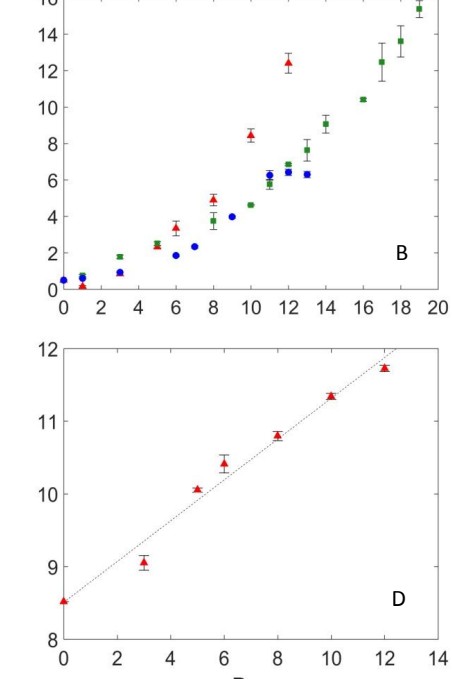









*Figure 2*. The evolution of cell density with time in culture experiments with *E. huxleyi* strain RCC911 (A:

high irradiance; B: low irradiance) and cell density on a logarithmic scale for nutrient-replete cultures (C:

high irradiance; D: low irradiance).






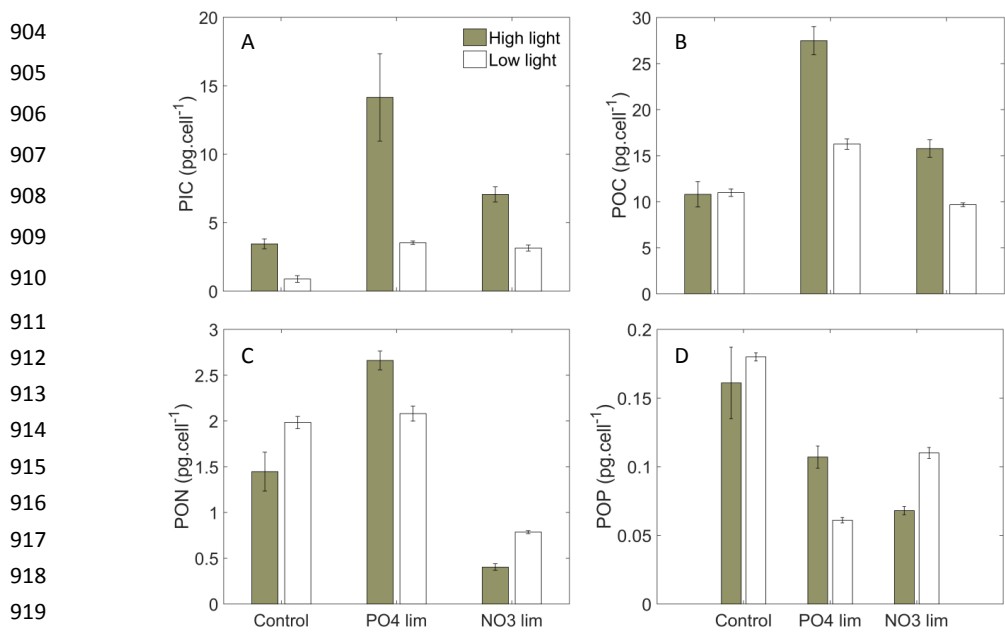

*Figure 3*. Cellular PIC, POC, PON, POP quotas.

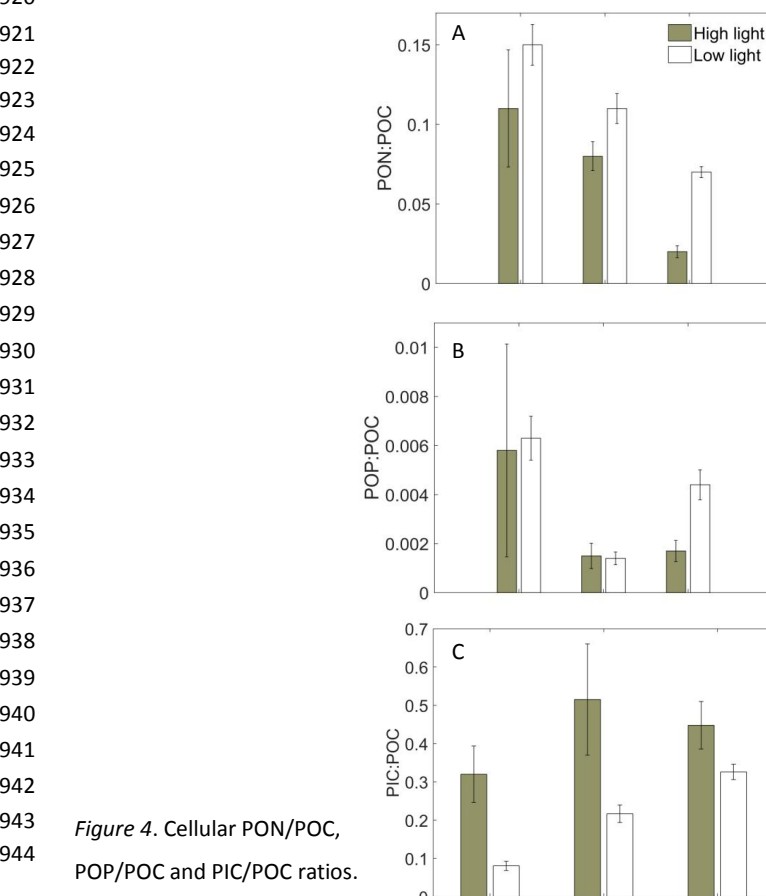

*Figure 4*. Cellular PON/POC,

POP/POC and PIC/POC ratios.



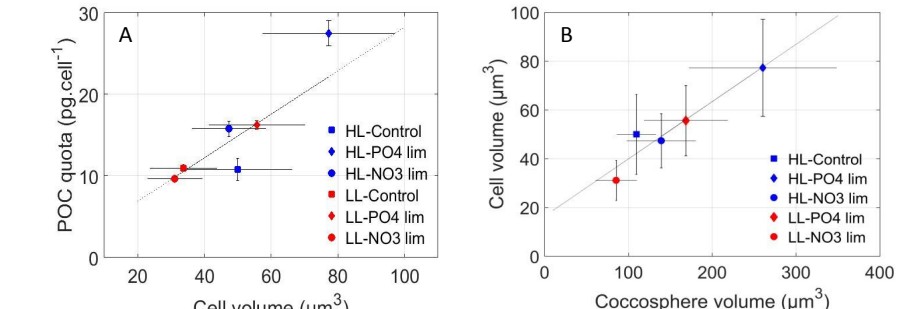

*Figure 5*. A: POC quota versus cell volume and B: cell volume against coccosphere volume in high light (HL) and low light conditions (LL).

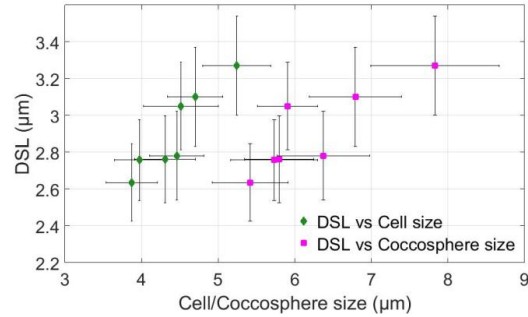

*Figure 6*. Distal shield length (DSL) versus coccosphere and cell diameter.

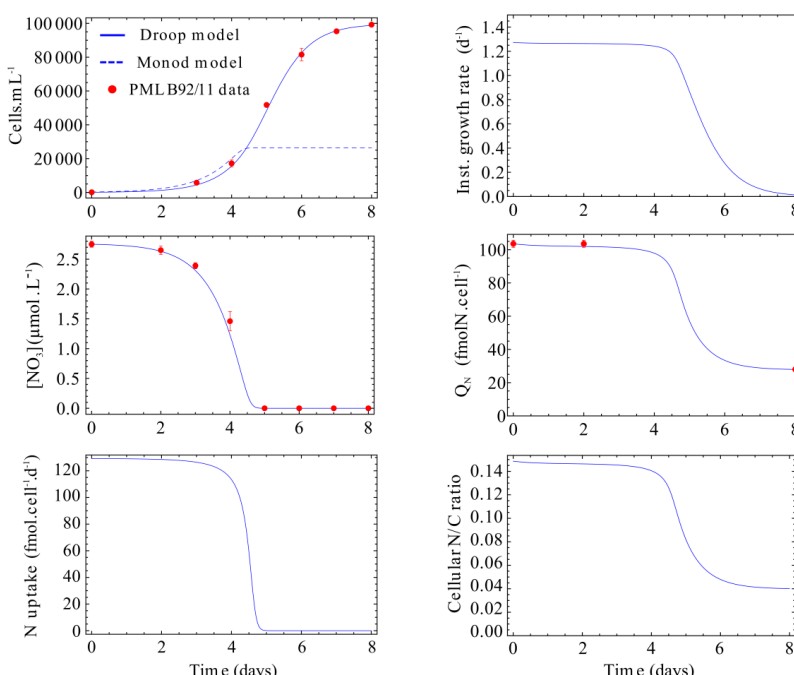

*Figure 7*. Model fitted to the data of the nitrate-limited cultures of Langer et al. (2013) (Inst = instantaneous)




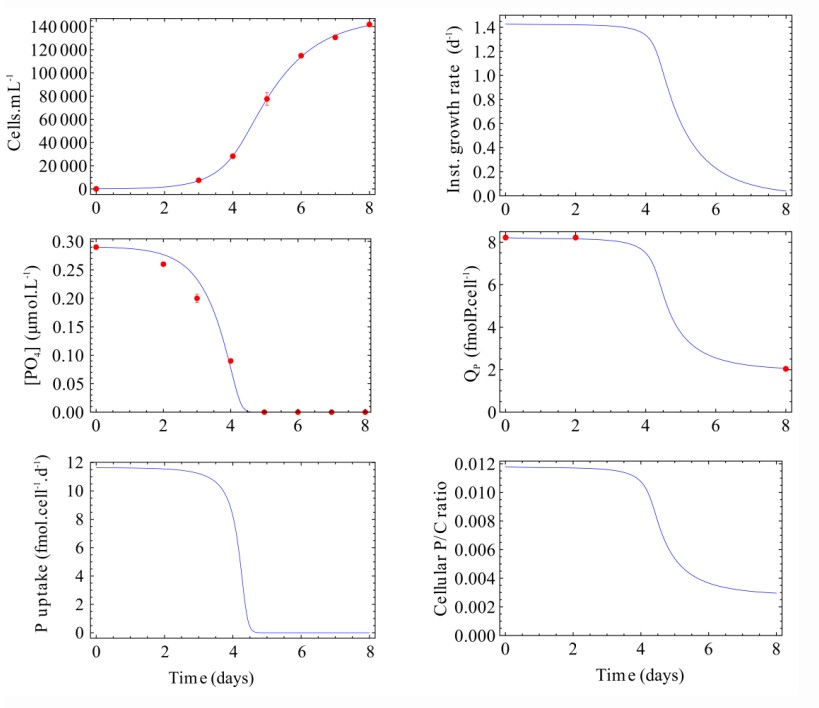


*Figure 8.* Model fitted to the data of the phosphate-limited cultures of Langer et al. (2013).

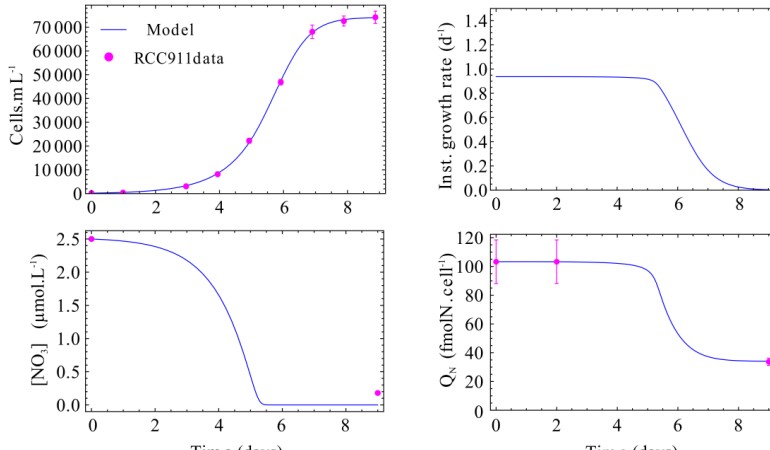



*Figure 9.* Model fitted to the data of the nitrate-limited cultures of strain RCC911 in high light conditions.







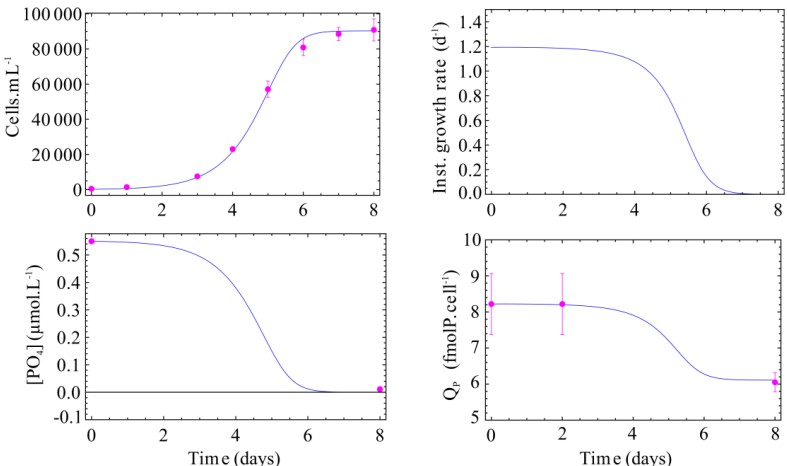


*Figure 10.* Model fitted to the data of the phosphate-limited cultures of strain RCC911 in high light conditions.



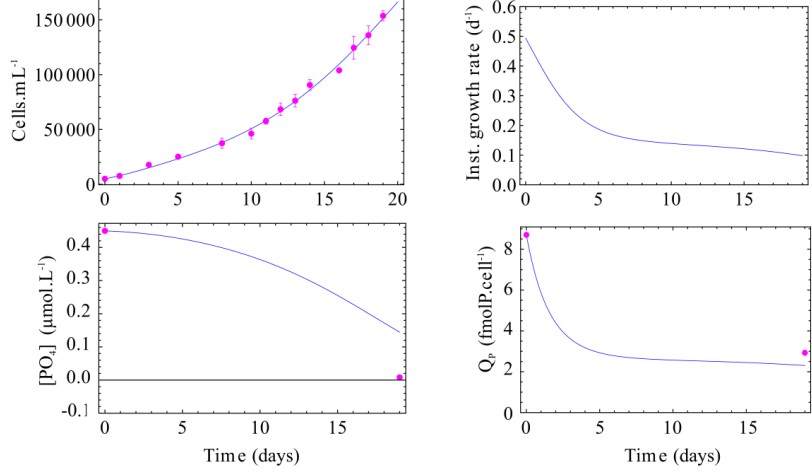


*Figure 11.* Model fitted to the data of the phosphate-limited cultures of strain RCC911 in low light conditions.










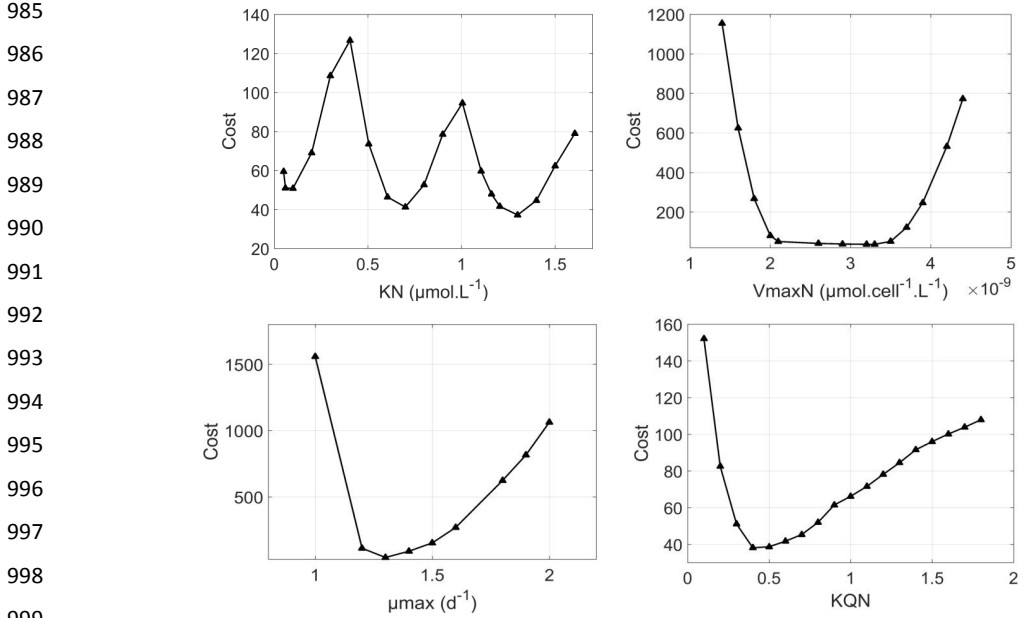

*Figure 12*. Variability of KN, VmaxN, µmax et KQN for the Langer et al. (2013) PML B92/11 experiment in nitrate-limited conditions.

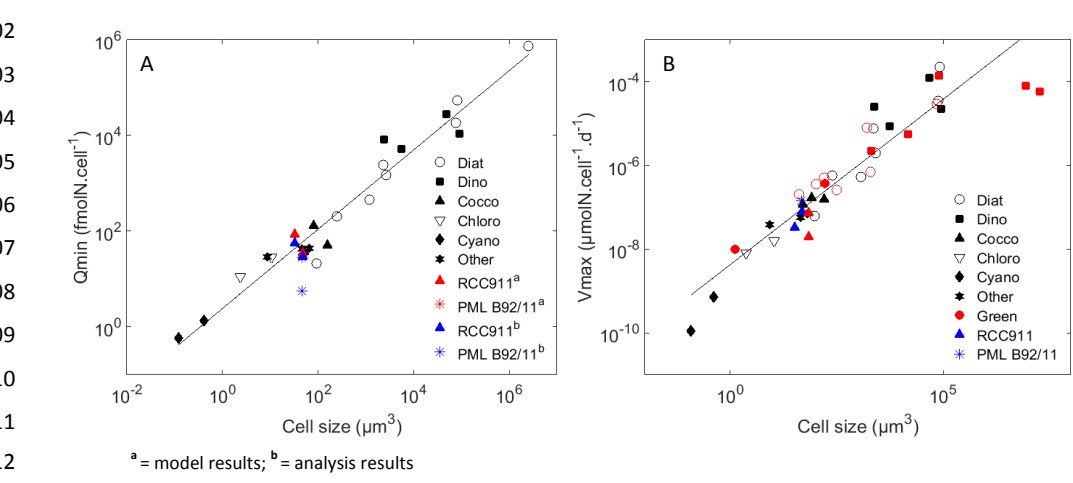

[a] = model results; [b] = analysis results

*Figure 13*. A) Minimum cellular quota $Q_{min}$ for nitrate versus the cell volume. Data of Maranon et al. (2013) and the results of the present study (model simulation results in red and analytical results in blue. B) Maximum normalized surface uptake $V_{maxN}$ for nitrate versus the cell volume. Data from Maranon et al. (2013) and Litchman et al. (2007) and the Droop model output for the experiments presented in this work.





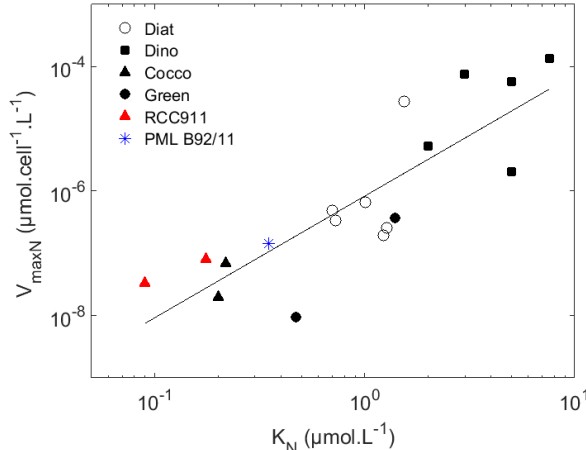


*Figure 14.* Data of Litchman et al. (2007) and results from the Droop model for RCC911 and PML B92/11
experiments in nitrate-limited conditions.



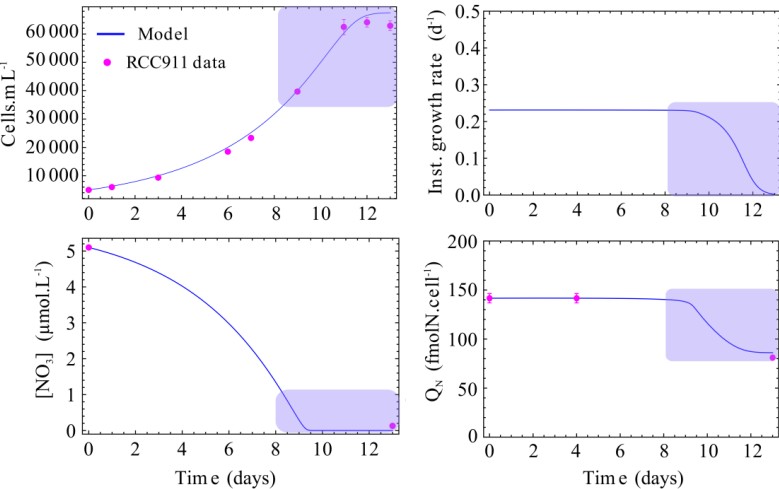



*Figure 15.* Model fitted to the data of the nitrate limited cultures on RCC911 strain in low light. The patch
corresponds to the equivalent nitrate concentration in the BIOSOPE ecological niche of coccolithophores at
the GYR station (between 150 and 200 m depth).















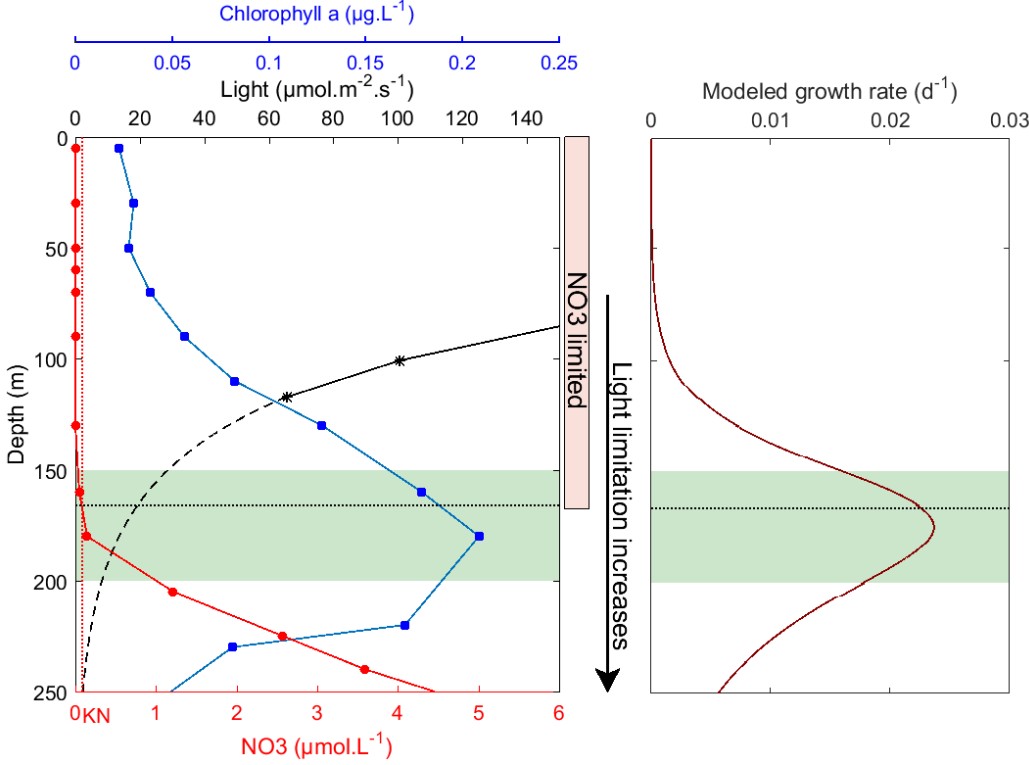



*Figure 16.* Left panel: In situ data (0 to 250 m) at the GYR station of the BIOSOPE transect (114.01° W, 26.06° S). Profiles of in situ measured chlorophyll a, PAR irradiance and nitrate concentration are shown. The dashed line represents an extrapolation of the irradiance between 117 m (last point measured) and 250 m considering a constant attenuation coefficient $K_d$ ($K_d$=0.025 m$^{-1}$ from Claustre et al., 2008) and a simple light calculation taken from MacIntyre et al. (2002). Dotted red line is the value of $K_N$ calculated with the Droop model and dotted black line is the depth at which this $K_N$ is observed. This depth also indicates the end of the nitrate limited part. Light limitation starts above the DCM and intensifies with depth. The green patch corresponds to the location of the maximum of coccosphere abundance taken from Beaufort et al. (2007) between 120° W and 107° W. The right panel shows the growth rate of *E. huxleyi* with depth at the GYR station (calculated using Eq. A8).





Table 1 : Growth rate, nutrient concentration, pH, DIC at the end of the experiments and shift in DIC
compared with the initial DIC (averages from triplicate, n=3 for growth rates and nutrients analysis).

| Sample | Growth rate[a] | std | NO3 | std | PO4 | Std | pH | std | DIC | std | DIC shift |
|---|---|---|---|---|---|---|---|---|---|---|---|
| | $d^{-1}$ | | $\mu mol.L^{-1}$ | | $\mu mol.L^{-1}$ | | | | $\mu mol.kg^{-1}$ | | % |
| **High light** | | | | | | | | | | | |
| Control | 0,91 | 0,03 | 67,92 | 1,98 | 3,95 | 0,12 | 8,13 | 0,01 | 2177 | 19,14 | 2,1 |
| PO4 lim | 0,00 | | 80,88 | 0,35 | 0,01 | 0,00 | 8,21 | 0,01 | 1894 | 21,01 | 12,1 |
| NO3 lim | 0,00 | | 0,18 | 0,03 | 5,74 | 0,00 | 8,14 | 0,00 | 2060 | 3,61 | 4,7 |
| **Low light** | | | | | | | | | | | |
| Control | 0,28 | 0,01 | 79,10 | 1,15 | 4,90 | 0,04 | 8,13 | 0,02 | 2161 | 7,55 | 4,1 |
| PO4 lim | 0,13 | 0,01 | 75,25 | 1,24 | 0,01 | 0,01 | 8,30 | 0,01 | 1956 | 8,33 | 13,2 |
| NO3 lim | 0,00 | | 0,13 | 0,02 | 5,83 | 0,02 | 8,09 | 0,00 | 2139 | 4,16 | 3,9 |

[a] = cells are in exponential growth phase at the end of control experiments

Table 2 : Cellular carbon, nitrogen and phosphorus quotas (averages from triplicate; n=6 for cellular
quotas measurements).

| Sample | PIC | std | POC | std | PON | std | POP | std | PIC:POC | std | PON:POC | std | POP:POC | std |
|---|---|---|---|---|---|---|---|---|---|---|---|---|---|---|
| | $pg.cell^{-1}$ | | $pg.cell^{-1}$ | | $pg.cell^{-1}$ | | $pg.cell^{-1}$ | | | | | | | |
| **High light** | | | | | | | | | | | | | | |
| Control | 3,46 | 0,36 | 10,80 | 1,38 | 1,45 | 0,21 | 0,16 | 0,03 | 0,32 | 0,07 | 0,11 | 0,04 | 0,0058 | 0,0043 |
| PO4 lim | 14,16 | 3,19 | 27,49 | 1,53 | 2,66 | 0,10 | 0,11 | 0,01 | 0,52 | 0,14 | 0,08 | 0,01 | 0,0015 | 0,0005 |
| NO3 lim | 7,06 | 0,55 | 15,77 | 0,95 | 0,40 | 0,04 | 0,07 | 0,00 | 0,45 | 0,06 | 0,02 | 0,00 | 0,0017 | 0,0004 |
| **Low light** | | | | | | | | | | | | | | |
| Control | 0,89 | 0,10 | 10,98 | 0,41 | 1,98 | 0,07 | 0,18 | 0,00 | 0,08 | 0,01 | 0,15 | 0,01 | 0,0063 | 0,0009 |
| PO4 lim | 3,53 | 0,25 | 16,25 | 0,56 | 2,08 | 0,08 | 0,06 | 0,00 | 0,22 | 0,02 | 0,11 | 0,01 | 0,0014 | 0,0003 |
| NO3 lim | 3,15 | 0,13 | 9,67 | 0,21 | 0,79 | 0,02 | 0,11 | 0,00 | 0,33 | 0,02 | 0,07 | 0,00 | 0,0044 | 0,0006 |


Table 3 : Cell, coccosphere volume and DSL (n=300 for coccosphere/cell measurements and n=100 for
coccoliths measurements) at the end of our experiments. No measurement of coccosphere and DSL
for control experiment in low light.

| Sample | Cell volume | | Coccosphere volume | | DSL | |
|---|---|---|---|---|---|---|
| | $\mu m^3$ | std | $\mu m^3$ | std | $\mu m$ | std |
| **High light** | | | | | | |
| Control | 49,97 | 16,38 | 109,5 | 23,3 | 3,05 | 0,24 |
| PO4 lim | 77,21 | 19,89 | 260,5 | 88,2 | 3,27 | 0,27 |
| NO3 lim | 47,33 | 11,13 | 139,2 | 41,2 | 2,78 | 0,24 |
| **Low light** | | | | | | |
| Control | 33,69 | 10,09 | | | | |
| PO4 lim | 55,64 | 14,42 | 168,6 | 50,0 | 3,10 | 0,27 |
| NO3 lim | 31,09 | 8,25 | 85,4 | 24,7 | 2,64 | 0,21 |




Table 4: Value of $Q_{N/P}^{min}$ (which corresponds to the cellular PON (POP) at the end of the experiment:
values measured and calculated) and the parameters obtained with the best-fit indicated for N and P
limited experiment (high light: HL and low light: LL).

| Strain | Light | Limitation | $Q_{N/P}^{min}$ | | Best-fit | | | |
| | | | Analysis $fmol.cell^{-1}$ | Calculation $fmol.cell^{-1}$ | $V_{maxN/P}$ $\mu mol.cell^{-1}.d^{-1}$ | $K_{N/P}$ $\mu mol.L^{-1}$ | $\mu_{max}$ $d^{-1}$ | $KQ_{N/P}$ |
|---|---|---|---|---|---|---|---|---|
| PML B92/11 | | $NO_3$ | 5,71 | 27,7 | $1,46.10^{-7}$ | 0,35 | 1,3 | 0,39 |
| PML B92/11 | | $PO_4$ | 1,935 | 2,04 | $1,37.10^{-8}$ | 0,051 | 1,57 | 0,98 |
| RCC911 | HL | $NO_3$ | 28,57 | 31,28 | $8,02.10^{-8}$ | 0,175 | 1 | 0,215 |
| RCC911 | HL | $PO_4$ | 3,464 | 5,931 | $1,86.10^{-8}$ | 0,49 | 1,6 | 1 |
| RCC911 | LL | $NO_3$ | 56,14 | 78,99 | $3,34.10^{-8}$ | 0,09 | 0,2 | 0,3 |
| RCC911 | LL | $PO_4$ | 1,968 | 2,875 | $7,43.10^{-10}$ | 0,45 | 0,5 | 0,45 |



Table 5 : In situ environmental conditions at the Deep ecological niche at 200 m depth at the GYR
station and initial conditions of the nutrient and light-limited experiment presented in this study.

| | BIOSOPE | RCC911 exp |
|---|---|---|
| T (°C) | 17,5-20 | 20 |
| Light ($\mu mol.m^{-2}.s^{-1}$) | < 20 | 30 (*Low light*) |
| $pCO_2$ (µatm) | ~ 400 | ~ 400 |
| $NO_3$ (µM) | ~ 1 | ~ 3 |
| $PO_4$ (µM) | ~ 0,2 | ~ 0,4 |
