# Peer review of "Growth of the coccolithophore *Emiliania huxleyi* in light- and nutrient-limited batch reactors: relevance for the BIOSOPE deep ecological niche of coccolithophores"

_Biogeosciences, 2016_

## Referee Comment (RC1) · Anonymous Referee #1 · 13 Jul 2016

Review of 'Growth of the coccolithophore Emiliania huxleyi in light- and nutrient-limited batch reactors: relevance for the BIOSOPE deep ecological niche of coccolithophores' by Perrin et al..

The manuscript addresses the deep ecological niche of coccolithophores found in nutrient poor oceanic gyres (like in the South Pacific) and the environmental conditions that control coccolithophore growth in this region. The study uses results from laboratory experiments and a model approach to define the growth response of E. huxleyi in regard to nitrogen, phosphorus and light limitation. Emiliania huxleyi represents one of

the most studied phytoplankton species and the obtained results from the conducted laboratory experiments are generally confirmed by the existing literature. The model approach, however, is a neat approach to gain a better understanding of the dynamical growth response of E. huxleyi when transitioning from nutrient replete exponential growth to nutrient (or light) limited growth and to translate these findings to natural phenomena.

Personally, I was excited to review the presented manuscript, especially as the abstract is well written and clear. This positive first impression, however, was depressed after a complete evaluation of the manuscript. In general I have the impression that this manuscript did not go through co-author review. The manuscript would greatly improve by a complete re-evaluation of the scientific presentation style. Below I will provide comments and suggestions to convince the editor that this manuscript does not meet the scientific standards of Biogeosciences and to encourage the author to consider a new submission of the manuscript after a comprehensive revision.

Main comments:

1) The manuscript has currently 16 figures which is overwhelming and in my opinion not necessary. It distracts the reader from the essential message. I encourage the authors to think about merging certain figures (e.g Fig. 7 - 11). Also, some data are presented as figure and table. Maybe some tables could be moved to supplementary materials.

2) As previously mentioned, E. huxleyi is one of the most studies phytoplankton species and I assume that additional literature (besides Langer et al. 2013) data will be available to test the applied model.

3) The language of the manuscript needs to be improved. Some sentences are very long and thus hard to understand. Some expressions are not correct or I cannot follow the line of thought (see some examples below). In general many parts of the text could be condensed. For example, the sections 4.1.1 to 4.1.3 are only repeating the results

previously mentioned and do not contribute to the discussion of the manuscript.

Additional comments:

1) Remove the period between units in the manuscript.

2) Use consistent colour and marker coding in all figures.

3) Line 18: The expression 'coccolithophore ecosystem' is not appropriate here. Maybe change to 'ecosystem for coccolithophores'.

4) L. 22-24: Avoid using 'physiology' and 'physiological' repetitive in one sentence.

5) L. 40-41 vs L. 44-45: Repetitive explanation of the connection between coccolithophore productivity and the influence on ocean-atmosphere CO2 fluxes.

6) L. 113: The use of K/2 media in regard to nitrogen limitation studies may provide problems because both NH4 and NO3 are present as nitrogen source. Therefore, it is possible that cells switch from NH4 uptake to NO3 uptake when NH4 is exhausted. This should be discussed.

7) L. 115: As cells were grown under batch conditions and allowed to enter the stationary phase (nutrient limitation), there is no possibility that cells are acclimated to the experimental conditions (of which nutrient limitation is on of the main factors).

8) L. 120: Light intensity is commonly expressed in 'umol quanta m-2 s-1'.

9) L. 140: At what point in time where the samples taken? Was it at the same time point for all treatments? When nutrient limitation starts in batch experiments the cellular physiology is changing in time as cells become nutrient starved. Please specify more clearly at what time point samples for volume, POC, PIC, etc. were taken.

10) L. 149: Here it is mentioned that coccolith width was measured but no data is presented.

11) L. 151: Please give an error estimate for the determination of the pH.

12) L. 167: The conversion from particulate Ca to particulate organic carbon should be stated.

13) L. 193: Why was the C/N ratio after Redfield used when the actual C/N ratio was analysed in the laboratory experiments?

14) L. 199: The half saturation constant K_n is sometimes used for K_nutrient and sometimes for K_nitrogen. This will be very confusing for a non-expert reader. Later it is called 'K_n/p'?

15) L. 210: Change 'Q_N^min' to 'Q_N^max'.

16) L. 277: If the stationary phase was not reached in the low light-Plim treatment, how can you be certain that the population was truly phosphorus limited? Is there the possibility that organic phosphate were available? E. huxleyi is known to be able to take up various forms of organic phosphates when inorganic PO4 is limited.

17) L. 294: What is meant by 'were harder to discern due to large error bar in high light and nutrient-replete conditions'?

18) L. 379: The expression 'ecologically dominant coccolithophore' is not necessarily correct for E. huxleyi. Maybe change to 'numerical dominant coccolithophore'.

19) L. 381-382: Leonardos and Geider (2005) used a non-calcifying strain of E. huxleyi thus it is quite obvious that they did not measure PIC per cell or the PIC:POC ratio.

20) L. 387: Previously a DIC consumption of 12-13 % is mentioned.

21) L. 396-399: Not clear, please rephrase.

22) L. 452-458: Very long sentence, please split into several.

23) L. 469: 'Nutrient limitation as well...' . What is meant by 'as well'?

24) L. 527-529: I cannot follow the line of thought.

25) L. 529-531: This is contradicting to the general observation of E. huxleyi blooms in

well stratified and sunlight ocean layers.

26) L. 550-554: This sentence is confusing. How is the reported observations of nutrient concentrations related to the phosphate limited experiment?

27) L. 557-559: Needs language check.

28) L. 572-574: Maybe the author want to mention and discuss the possibility of grazing and vertical export.

29) L. 585-590: Very long sentence.

30) L. 610-614: It is not really a new finding that batch experiments can be used to determine physiological parameters such as half-saturation constants and maximum uptake rates.

31) L. 679: Why is the BG Discussion paper of Beaufort et al. cited instead of the final and revised version?

32) Table 2: Why reporting PON:POC instead of the commonly reported POC:PON? This would facilitate a direct comparison to Redfield. Use decimal point instead comma.

There are many more points that need to be addressed. I only gave some examples and it cannot be the reviewers task to invest a considerable amount of time to discuss and mention all. I encourage the authors to invest the time to bring this manuscript in a more presentable format which will certainly facilitate a further review process.

---

## Referee Comment (RC2) · Anonymous Referee #2 · 19 Aug 2016

GENERAL COMMENTS The manuscript by Perrin et al differs from much of the physiological work in the coccolithophore literature in that it examines co-limitation (nutrients and light) and uses modelling to further parameterise key physiological parameters. It is generally well written, although at times it does become repetitive and parts could be tightened and shortened to avoid this, also there are a few grammar/syntax errors which distract from the work. 16 figures and 5 tables is excessive, especially when the same data appears in both and the authors should consider the best way to convey their results and key points without repeatedly presenting the same data. The authors also need to think over their central messages and conclusions from this work

(see comments below) which would lead to a better structure of the paper (e.g., what belongs as supplementary material or doesn't need to be emphasised repeatedly). Overall the scientific work is excellent and will provide significant new insights into E. huxleyi physiology which can be taken up by other studies.

1. Emiliania huxleyi, not all coccolithophores. Unavoidably, this is a central bugbear of much of the literature – information on E. huxleyi does not equal information on coccolithophores as a whole. E. huxleyi is just one species of a group which exists in a wide diversity of ecological niches. Although E. huxleyi is particularly cosmopolitan in its distribution and often becomes an exhibitionist when forming mesoscale sub-polar blooms, it is lightly calcified and contains little POC. Though E. huxleyi may dominate calcite fluxes in high-latitude environments, other (often deep-dwelling) species dominate fluxes to sediments below oligotrophic gyres. Please be specific when discussing coccolithophores as a group or E. huxleyi as a single species.

2. Why the deep niche focus? Despite the introduction, it is not clear why the focus on the deep E. huxleyi communities in the South Pacific. Co-limitation and the physiological parameters determined in this study are relevant to E. huxleyi growth in many other environments. Growth in this environment is likely to be taxing (and potentially on the outer envelope of the E. huxleyi global niche), but so is growing in the cold and dark, nutrient poor Arctic or Antarctic (where E. huxleyi also has sizable communities). The strength of the work is outlined in the introduction (Lns 79-80): 'understanding the development of (deep) coccolithophore populations in low nutrient, low irradiance environments does contribute to building a global picture of coccolithophore ecology and biogeography'. At times the authors seem overly fixated on the deep communities in the South Pacific – despite having used a surface water isolate potentially adapted to growth with low inorganic nutrients and readily available organic nutrients.

3. What are the main messages from this work? That batch cultures and modelling can determine good estimates of physiological parameters is a nice result, but is it the main message of the paper (and does it need repeating several times throughout the

manuscript?) What is the main message? The physiological parameters determined are fascinating and have potentially global implications (and don't appear to differ with other E. huxleyi strains). Did you need to look at a BIOSOPE strain for this work and are the results only relevant to the 'deep ecological niche' of coccolithophores. This issue over the main message of the paper partly results in the 16 figures and 5 tables and repetitive nature of parts of the manuscript (please note that much of the data in tables is in the figures and vice versa and consider deleting figures where the data is better shown in a table – note you don't have to show figures just to highlight statistical relationships). Could some of the modelling plots go in the Supplementary material?

4. Light dose. It should be acknowledged that light dose is important, not just the incidental amount of light. For example, a 12:12 L:D day at 30 umol photons m-2 s-1 may be similar to a 16:8 L:D day at 20 umol photons m-2 s-1. This should be made clear when comparing between in situ and laboratory, and also between various laboratory studies.

8. Alternative sources of N and P. Several papers (e.g., Benner & Passow (2010) Utilization of organic nutrients by coccolithophores. Marine Ecology Progress Series 404, 21-29) have shown that E. huxleyi can utilise organic sources of N and P, whereas the focus in this paper is on inorganic (nitrate, phosphate) sources only. In an oligotrophic environment, alternative sources of nutrition (organic substrates) will be especially important. The existence of these other nutritional strategies should be acknowledged and discussed – how reliant are the conclusions of this paper on the assumption that E. huxleyi is not utilising organic nutrient sources?

SPECIFIC COMMENTS Ln 18: What does 'coccolithophore ecosystem' mean? Revise use of ecosystems. Also, why are they important – production, export?

Ln 18-19: Did you actually 'investigate the conditions that regulate the development of a deep coccolithophore niche' or rather investigate the physiological responses of an E. huxleyi strain to living in simulated conditions of living deep in the oligotrophic

gyres?

Ln 30: How can an ecosystem be disadvantageous? I am sure the other members of the DCM community don't feel that it is disadvantageous.

Lns 38-39: Consider you use of 'participate' – a better word would be contribute.

Ln 40-41: The ratio of calcification and photosynthesis are not the only factors contributing to ocean-atmosphere CO2 fluxes: what about dissolution, respiration, advection, sinking, or carbonate chemistry? This sentence oversimplifies something far more complex than PIC:POC.

Ln 42-43: Sure all these factors influence cellular PIC:POC (in laboratory cultures) for coccolithophores, but not for the whole phytoplankton community (i.e. the communities POC). Consider being more specific in this opening paragraph to avoid confusion.

Ln 44: biogeography is not just growth rates, what about mortality?

Ln 47: This list of references is not extensive, but more examples of relevant literature (the list would be much longer) – hence start the list with "e.g.". See also the list on Ln 50.

Ln 60: Communities of deep living coccolithophores were not first discovered by BIOSOPE – the existence of deep shade flora has been known since the 1970's (at least). Rather, BIOSOPE observed it in the South Pacific Ocean. See also comments on the deep niche of coccolithophores.

Ln 62: Firstly, nitracline (nitrate) or nutricline (all nutrients) – do all nutrients have a 'cline' at 200 m (nitrate, phosphate, silicate, ammonia?)? Secondly, the nitracline is defined as the depth were nitrate concentrations increase over a threshold, it doesn't not have a fixed depth as is inferred in this line.

Ln 114: What is the rationale for using a surface E. huxleyi isolate to replicate the biology of a deep E. huxleyi community? Are they the same?

Lns 120-123: Surely daily light dose is more relevant to replicating in situ conditions that instantaneous light levels? 12 hours of 30 umol m-2 s-1 is not the same as 16 hours of 20 umol m-2 s-1 (or is it?).

Ln 159: Where nutrients analysed on a CHN Auto-analyzer? Or a Seal Analytical AAIII? I cannot find this equipment on the Seal website.

Ln 168-170: How was POP measured? Fumed or not fumed? It is not clear from this text, and as these are among the few measurements of cellular P for E. huxleyi it is quite important to state the method.

Ln 189: N fixation rate? Or N uptake rate?

Ln 192-194: Why use the cellular carbon quota and Redfield to calculate cellular N content? The N content has been measured and why assume Redfield?

Ln 202-205: Why has two methods used to determine cell volume and surface area? When was one used and not the other in the calculations?

Ln 214: PON and POP are not equivalent to cellular NO3 or PO4, or do you mean the media concentrations? This sentence is confusing.

Ln 216: What other internal cellular quota of N is there? What is nutrient N?

Ln 243: What is KN/P? N to P ratio? Same for QN/P in Ln 246.

Ln 250-251: Do you mean nutrient data points (i.e. media nutrient concentration) or nutrient-quota (i.e. cellular elemental quota)?

Ln 269,279,288,306: Do you need sub-headings for the sub-section of the results?

Ln 293: Please make it clear why you express as P:C and not C:P.

Ln 379-381: How do new insights into E. huxleyi physiology tell you about the other coccolithophores living deep in the water-column? Maybe other Isochrysidales or Noelaerhabdaceae, but not species such as Florisphaera profunda.

**BGD**

Ln 409: Rephrase 'however, large error margins do not allow explaining these observations ..'

Ln 442: Decreasing phosphate/nitrate or phosphate/nitrate-limitation? A drop in nutrients or their complete absence?

Ln 447: Please, the review by Zondervan (2007) is almost entirely E. huxleyi (maybe a little Gephyrocapsa), but it certainly isn't 'coccolithophores'.

Ln 452-460: Such a comparison between studies needs to account for day length differences between experiments – i.e. where the light doses different?

Ln 463-465: The relationship between coccolith size and coccosphere size is likely to be (very!) species-specific and hence this sentence should be rephrased. Also, PIC quota of what? Coccoliths yes, but not coccospheres – there is considerable intra- and inter-species variability in the number of coccoliths per cell.

Ln 467,476, 483: Having three separate summary sections mid-discussion breaks up the flow of the paper and doesn't seem necessary if they are combined and written clearly.

Ln 561: What other nitrate is there?

Ln 573: What about comparing these growth rates with other measurements/estimates? Figure 13B. What are the red triangles?

---

## Author Comment (AC1) · 15 Sep 2016

Dear Editors,

We have read and considered the comments made by reviewer 1 to our manuscript. We found these comments pertinent and think they will improve the manuscript. In the following we detail our responses to specific questions and are prepared to implement these corrections/changes should the article be accepted for publication in Biogeosciences.

Answers to main comments:

[Figure]

1) We agree with this comment and we will merge some figures and add a supplementary part for tables and/or figures that are not indispensable in the text. Figures 1, 8, 10, 11, 12 and table 3 will be placed in a supplementary material. Figures 5 and 6 will be merged such as figures 13 and 14. Table 5 will be removed and values will be described in the text of the manuscript.

2) As the reviewer points out, our physiological model can be applied to datasets other than the Langer et al. (2013) dataset and the dataset we obtained with our own batch experiments. In our opinion Langer et al. (2013) is the best dataset available for our modeling exercise because sampling for measurement of medium and cellular chemical composition was done with a high frequency in this experiment. We hope our approach will also be used with other datasets in the future. However, we chose not to include other datasets in the present manuscript because this would have significantly increased the length of the manuscript without adding much in terms of the new modeling method proposed. We think that the modeling we present is strong enough to support our conclusions on the environmental controls on E. huxleyi distribution in the deep ecological niche of South Pacific Gyre.

3) We will improve the language and make the text more concise. The sub-sections 4.1.1 to 4.1.3 will be deleted from the manuscript.

Answers to additional comments:

1) The change will be implemented as recommended.

2) The change will be implemented as recommended.

3) This will be changed to "a poorly known but potentially important ecosystem for coccolithophores".

4) The change will be implemented as recommended.

5) The change will be implemented as recommended.
6) It is true that we did not specify that we only added nitrate and phosphate to the medium and that we did not add the NH4Cl indicated in the reference medium in order to avoid the problem of multiple nitrogen sources. This will be explained in the revised text.

7) We agree with this comment and will state in the revised text that prior to experiments cells were only acclimated to temperature, light and carbon chemistry conditions, but not to low nutrients. Acclimation to low nutrient concentrations is not possible in a batch approach, because acclimation presupposes constant conditions. The central feature of a batch culture, by contrast, is that nutrient concentrations change over the course of the experiment.

8) The change will be implemented as recommended.

9) Samples were always taken in the afternoon between noon and 4pm, and always in the same sampling order. Each culture bottle sampling operation took approximately 45 min, preventing samples from different bottles from being taken at the same time. However, each analytic value was averaged over the three replicates. This will be specified.

10) This is correct and the statement about coccolith width measurements will be removed.

11) We will mention that the error for pH measurements is 0.02 pH units.

12) We will mention that the conversion from the concentration of Ca to particulate inorganic carbon (PIC) is based on a 1:1 stoichiometry between $Ca^{2+}$ and PIC. i.e. all the calcium in the filters is considered to originate from calcium carbonate (Fagerbakke et al., 1994).

13) It is true that we made a mistake on this point. The C/N ratio for the nitrate uptake calculation is not necessary for this calculation because the PON data are available for the control experiments. To correct this point we will change the Monod plot (Fig. 7)

and the part in the text that describes this point. This will entail only minor difference in the model results because the C/N ratios for the control experiments were near the Redfield ratio: for example the C/N ratio for the control NO3 experiment of Langer et al. (2013) was 5.72.

14) To clarify notation, we propose to use KR for nutrients in general, KN for the nitrate half-saturation constant and KP for the phosphate half-saturation constant.

15) The change will be implemented as recommended

16) This is an interesting comment. In fact, we would like to be less assertive in the revised manuscript with regards to P-limitation being attained in the low-light PO4-limited culture. We propose to discuss the following hypotheses:

(1) P-limitation was not attained in the low light and low PO4 experiment. The cells were P-limited. This can be inferred from a) the POP quota, which is lower than that of the control; b) the POP:POC ratio (POC:POP), which is also lower (higher) than that of the control; c) a deviation of the growth curve from exponential growth starting on day 16 (out of 19) at the very latest. While a decline in POP quota is an early sign of limitation, the decline in growth rate is a late appearing sign, indicating severe limitation. The cessation of cell division (stationary phase) is merely the last stage in the process of becoming P-limited over the course of a batch culture.

(2) Cells used another source of phosphorus such as organic phosphorus. However, no other sources of phosphorus other than the added phosphate were present in the culture medium except possible organic sources present in the initial seawater. If organic phosphorus sources were available in the medium, cells in the high light low PO4 experiment would not have been expected to reach the stationary phase. This hypothesis is not rejected but seems not to be the reason for this absence of a stationary phase.

17) We made a mistake in calculating the magnitude of the error bar which is in fact

smaller that we thought; we will thus remove this sentence.

18) The change will be implemented as recommended.

19) We mention the Leonardos and Geider (2005) experiment because it is the only experiment to our knowledge where nutrient-and light co-limitation was carried out. We will mention that this experiment was carried out with a non-calcifying strain.

20) We will change the sentence to " . . .to ensure that changes in the carbonate system were within a minimal range (< 10% except for the P-limited experiment where the DIC change were 12 and 13%; Table 1)".

21) This sentence will be rephrased according to comment 16.

22) The change will be implemented as recommended.

23) As mentioned in main comment 3, these sub-sections will be removed from the manuscript.

24) We will modify this sentence in order to be clearer: "According to Gregg and Casey (2007) the key to their success in the global oceans is to find areas where nutrients and light are low enough to inhibit growth of diatoms and chlorophytes, but where there is sufficient vertical mixing to prevent excessive their sinking losses or where they can find nutrients at depth under low illumination levels."

25) We agree that there is a contradiction and propose to modify the relevant sentence. In fact, in stratified, sunlit portions of the upper ocean, coccolithophore blooms occur after diatom blooms have depleted the nutrients. This advantage over diatoms in nutrient-depleted waters is a consequence of the greater affinity for nutrients of coccolithophores. Overall, coccolithophores have an advantage in low nutrient and low light compared to diatoms and chlorophytes (Balch, 2004; Gregg and Casey, 2007). This explains their development in the low-light, low-nutrient waters in some regions of the ocean (Beaufort et al., 2008; Haidar and Thierstein, 2001; Jordan and Winter, 2000).

26) This sentence will be change and was actually not clear. What we wanted to say is that phosphate measurements in the depth range 0-100 m in the GYR station of the BIOSOPE transect were always above 0.1 micromoles per litre, which suggests that PO4-limiting conditions are not attained in this water column (Moutin et al.,2008).

27) The sentence is confusing and will be simplified.

28) A short discussion of grazing and vertical export as parameters that contribute to defining the distribution of coccolithophore populations will be included in the revised manuscript.

29) The sentence in question will be split into two.

30) We need to modify this sentence. While it is not possible to obtain reliable half-saturation constants for nutrient uptake in a batch experiment (a chemostat experiment is necessary), other parameters such as the maximum growth rates and maximum uptake rates can indeed be estimated in a batch experiment. As far as we know the only literature found to estimate the half-saturation constant for nutrient uptake for E. huxleyi using a batch culture is from Eppley et al. (1969). However, we think that the transient character of batch cultures makes the determination of half-saturation constants very difficult. We propose to circumvent this difficulty by modeling the batch experiments with a simple Droop model that enables us to extract information on nutrient affinity (the half saturation constant) from the transient results of the batch experiment

31) Reference to the final revised version of Beaufort et al. (2008) will be made.

32) POC: PON and POC:POP will be reported and the decimal point will be used instead of the comma in the final manuscript.

References

Balch, W. M.: Re-evaluation of the physiological ecology of coccolithophores, in Coccolithophores, edited by P. D. H. R. Thierstein and D. J. R. Young, pp. 165–190, Springer Berlin Heidelberg., 2004.

Beaufort, L., Couapel, M., Buchet, N., Claustre, H. and Goyet, C.: Calcite production by coccolithophores in the south east Pacific Ocean, Biogeosciences, 5, 1101–1117, 2008.

Eppley, R. W., Rogers, J. N. and McCarthy, J. J.: Half-Saturation Constants for Uptake of Nitrate and Ammonium by Marine Phytoplankton, Limnol. Oceanogr., 14(6), 912–920, doi:10.4319/lo.1969.14.6.0912, 1969.

Fagerbakke, K. M., Heldal, M., Norland, S., Heimdal, B. R. and Båtvik, H.: Emiliania huxleyi. Chemical composition and size of coccoliths from enclosure experiments and a Norwegian fjord, Sarsia, 79(4), 349–355, doi:10.1080/00364827.1994.10413566, 1994.

Gregg, W. W. and Casey, N. W.: Modeling coccolithophores in the global oceans, Deep Sea Res. Part II Top. Stud. Oceanogr., 54(5–7), 447–477, doi:10.1016/j.dsr2.2006.12.007, 2007.

Haidar, A. T. and Thierstein, H. R.: Coccolithophore dynamics off Bermuda (N. Atlantic), Deep Sea Res. Part II Top. Stud. Oceanogr., 48(8–9), 1925–1956, doi:10.1016/S0967-0645(00)00169-7, 2001.

Jordan, R. W. and Winter, A.: Assemblages of coccolithophorids and other living microplankton off the coast of Puerto Rico during January–May 1995, Mar. Micropaleontol., 39(1–4), 113–130, doi:10.1016/S0377-8398(00)00017-7, 2000.

Langer, G., Oetjen, K. and Brenneis, T.: Coccolithophores do not increase particulate carbon production under nutrient limitation: A case study using Emiliania huxleyi (PML B92/11), J. Exp. Mar. Biol. Ecol., 443, 155–161, doi:10.1016/j.jembe.2013.02.040, 2013.

Litchman, E., Klausmeier, C. A., Schofield, O. M. and Falkowski, P. G.: The role of functional traits and trade-offs in structuring phytoplankton communities: scaling from cellular to ecosystem level, Ecol. Lett., 10(12), 1170–1181, doi:10.1111/j.1461-

0248.2007.01117.x, 2007.

Marañón, E., Cermeño, P., López-Sandoval, D. C., Rodríguez-Ramos, T., Sobrino, C., Huete-Ortega, M., Blanco, J. M. and Rodríguez, J.: Unimodal size scaling of phytoplankton growth and the size dependence of nutrient uptake and use, Ecol. Lett., 16(3), 371–379, doi:10.1111/ele.12052, 2013.

Riegman, R., Stolte, W., Noordeloos, A. A. M. and Slezak, D.: Nutrient uptake and alkaline phosphatase (ec 3:1:3:1) activity of Emiliania huxleyi (PRYMNESIOPHYCEAE) during growth under N and P limitation in continuous cultures, J. Phycol., 36(1), 87–96, doi:10.1046/j.1529-8817.2000.99023.x, 2000.

---

## Author Comment (AC2) · 15 Sep 2016

Dear Editors,

We have read and considered the comments made by reviewer 2 to our manuscript. We found these comments pertinent and think they will improve the manuscript. In the following we detail our responses to specific questions and are prepared to implement these corrections/changes should the article be accepted for publication in Biogeosciences. In particular, the overall clarity of the manuscript will be improved, including shortening the text, improving grammar/syntax, and decreasing the number of figures

and tables in the main text by moving part of them to the supplementary material. Figures 1, 8, 10, 11, 12 and table 3 will be placed in a supplementary material. Figures 5 and 6 will be merged such as figures 13 and 14. Table 5 will be removed and only values will be described in the text of the manuscript.

General comments:

1) Emiliania huxleyi and not coccolithophores in general. We have made changes throughout the text to avoid referring to E. huxleyi as a model for coccolithophores in general.

2) The deep niche focus was chosen for two reasons: (1) little is known about E. huxleyi growth in these low-nutrient, low-light conditions despite the fact that they could represent a non-negligible portion of the global E. huxleyi population, and; (2) the BIOSOPE transect is unique in the breadth of physical and chemical parameters measured, which makes our joint experimental/modeling exercise easier. We do acknowledge, however, that we might have missed the relevance of our work for other environments including the cold, dark and nutrient-poor Arctic and Antarctic. We will explain the relevance of our findings to environments other than the deep ecological niche in a revised manuscript.

3) The main message of our work is that batch experiments coupled to simple physiological modeling can help interpret environmental controls on distributions of coccolithophore populations in the ocean. We agree that this needs clarification and the subordination of the BIOSOPE deep niche approach to this overall message needs to be better stated in the manuscript. Once this message is delivered more clearly than in the original submission, we have to stress that the BIOSOPE deep niche was chosen for the reasons explained above. The deep niche study chosen to apply our approach is the best possible field situation based on the available published datasets of chemical and physical properties. Figures and Tables will be reorganized as stated in the first paragraph of this reply.

4) We agree that depending on the light cycle, the amount of light available for the cell (light dose) will be different. This point and the specification of the L:D cycle will be added to the text and discussed in addition to the intensity of irradiance.

8) We agree that the possibility of organic nutrient utilization needs to be discussed with reference to the oligotrophic environment of the South Pacific Gyre. What is interesting is that the physiological parameters constrained by our model, which considers only inorganic nutrients, result in a depth of the potential growth rate that coincides with that DCM and the maximum number of coccolithophore cell counts in the GYR station of the BIOSOPE transect. We therefore conclude that either organic nutrients are not used significantly, or if they are used, the pattern of their distribution in the water column mimics that of inorganic nutrients. We cannot exclude this second possibility although from the correspondence between the modeling and the in situ vertical distribution of coccolithophore cells and chlorophyll, it is very likely that inorganic nutrients play the predominant role in controlling the vertical position of the coccolithophores. This will be added to the discussion section.

Specific comments:

Ln 18: The word "ecosystem" is not used in a correct way. We will change this part of the sentence and add the reason why coccolithophores are important for organic carbon and mineral export.

Ln 18-19: We will change this sentence as recommended.

Ln 30: We will change this sentence to "...metabolism and behavior in a low light and low nutrients environment of the ocean".

Ln 38-39: This will be changed as recommended.

Ln 40-41: We will briefly mention the other factors that influence the ocean-atmosphere $CO_2$ fluxes.

Ln 42-43: We will be more specific in saying that the mentioned factors influence the

PIC:POC ratio of coccolithophores but not of the whole phytoplankton community.

Ln 44: This will be specified in this sentence.

Ln 47: The sentence will be added as recommended.

Ln 60: The term "discovered" will be changed to "observed". Deep photic zone (low light) communities of coccolithophores have been observed in the North and Central Pacific at least since the work of Okada and Honjo (1973).

Ln 62: We agree that the sentence is not correct and needs to be changed. Nitrate and phosphate have actually their 'clines' around the same depth throughout the transect. We need to specify that the nitracline and the phosphacline were observed around 200 m at the GYR station (in the middle of the South Pacific Gyre), but of course this depth is not the same along the transect as both nitracline and phosphacline are shallower at the extremities of the gyre.

Ln 114: We chose to work with a surface strain from the BIOSOPE transect because no E. huxleyi strains were isolated inside the gyre at 200 m depth. This is a limitation of our study that we will mention.

Ln 120-123: A model of the PAR daily cycle at the date and the coordinates of the GYR station was used to calculate the L:D cycle (Figure 1). This was between 14:10 and 12:12 along the whole transect. Thus, the 12:12 cycle used in our experiments is representative of the in situ situation. This point will be specified in the manuscript.

Ln 159: Nutrients were measured on a Seal Analytical auto-analyzer model AA3. Here is the relevant link: http://www.seal-analytical.com/Products/AA3HRAutoAnalyzer/tabid/59/language/en-US/Default.aspx

Ln 168-170: POP was measured as the difference between the total particulate phosphorus and the particulate inorganic phosphorus. Both were analyzed using a Seal Analytical 3 auto-analyzer after some different analytical steps summarized in the paper of Labry et al. (2013). At one step filters were hydrolyzed using a HCl solution, so

filters were not fumed. We will specify these points in the manuscript.

Ln 189: It is N-uptake. The relevant symbol will be changed to Nup to be in accordance with the Droop model. However, nutrient uptake and nutrient fixation are equivalents in the Monod theory because nutrients are assimilated as soon as they are taken up.

Ln 192-194: True, as noticed by the first reviewer as well, we actually made an error on this point. The C/N ratio for the nitrate uptake calculation is not necessary for this calculation because the PON data are available for the control experiments. To correct this point we will change the Monod plot (Fig. 7 in the manuscript) and the text that describes this point. This will entail only a minor difference in the model results because the C/N ratios for the control experiments are near the Redfield ratio: for example the C/N ratio for the control NO3 experiment of Langer et al. (2013) was 5.72.

Ln 202-205: We need to be clearer about these two different methods to determine cell volume and surface area. Cell volume (or surface area) were calculated for the experiment of Langer et al. (2013) because of the lack of measurements, while cell volume was directly measured in the experiment presented in this manuscript and reported in the "experimental" part of the "Materials and methods".

Ln 214: We will change "NO3 and PO4" to "N and P" to avoid confusion.

Ln 216: We will change the nutrients notation in the text because of the existing confusion between nutrient N and nitrogen N. We will refer to nutrients in general with the letter R, to the nutrient nitrogen with the letter N and to nutrient phosphate with the letter P.

Ln 243: (see answer to previous comment). We will change the notation for the half saturation constants for nutrient uptake: KN is the constant for nitrate uptake, KP is the constant for phosphate uptake and KR is the generalized constant for nutrient uptake. Same thing for the nutrient quotas, e.g. QN/P, that will be referred to as QR.

Ln 250-251: We mean nutrient cellular quota.

Ln 269, 279, 288, 306: We will remove these sub-headings.

Ln 293: We will express ratios as C:P and C:N when revising the manuscript

Ln 379-381: True and as specified in the general comments of this review, we need to be more careful when we talk about E. huxleyi and coccolithophores as a group. Of course this work gives us new insights for the species E. huxleyi and maybe for other Isochrysidales or Noelaerhabdaceae but undoubtedly not for all coccolithophore species.

Ln 409: This sentence will be rephrased.

Ln 442: We will change "for decreasing phosphate than for decreasing nitrate" to " for phosphate limitation than for nitrate limitation".

Ln 447: We will modify this sentence making it clear that Zondervan (2007) is almost entirely based on E. huxleyi results.

Ln 452-460: As in general comment, the light dose will be added to the text in order to improve the comparison and because of the importance of the light dose and not only the light intensity. Only Feng et al. (2008) used a 12:12 L:D cycle, but the other mentioned studies Rokitta and Rost (2012), Trimborn et al., (2007) and Zondervan et al. (2002) used a 16:8 L:D cycle. We will change this paragraph to be more specific and avoid comparing experiments with very different L:D cycle experiments.

Ln 463-465: We will rephrase this sentence as recommended to be clearer on the species-specific relation between coccolith size and coccosphere size and to take into account the fact that the PIC quota per coccolith could be estimated by the size of coccoliths but that the PIC per coccosphere depends on the number of coccoliths per cell.

Ln 467, 476, 483: These sub-sections will be deleted to avoid repetition of the discussion.

Ln 561: This sentence will be rephrased. Other sources of nitrogen might include organic nitrogen, although based on the modeling results (see answer to comment 8, above) we think that inorganic nitrogen dominates over organic nitrogen.

Ln 573: We will add some comparisons and references in the revised text, especially papers of Laws (2013) and Selph et al. (2011) which evaluate and estimate respectively in situ growth rates considering the mortality of phytoplankton due to grazing. Consequently, their estimation of growth rate will be lower than the net growth rate ad need to be compared carefully with our estimation. In the legend of figure 13B, the red triangles are the coccolithophore data from Litchman et al. (2007) (black points are the data of Marañón et al., 2013). This will be specified in the legend.

References

Feng, Y., Warner, M. E., Zhang, Y., Sun, J., Fu, F.-X., Rose, J. M. and Hutchins, D. A.: Interactive effects of increased pCO2, temperature and irradiance on the marine coccolithophore Emiliania huxleyi (Prymnesiophyceae), Eur. J. Phycol., 43(1), 87–98, doi:10.1080/09670260701664674, 2008.

Labry, C., Youenou, A., Delmas, D. and Michelon, P.: Addressing the measurement of particulate organic and inorganic phosphorus in estuarine and coastal waters, Cont. Shelf Res., 60, 28–37, doi:10.1016/j.csr.2013.04.019, 2013.

Langer, G., Oetjen, K. and Brenneis, T.: Coccolithophores do not increase particulate carbon production under nutrient limitation: A case study using Emiliania huxleyi (PML B92/11), J. Exp. Mar. Biol. Ecol., 443, 155–161, doi:10.1016/j.jembe.2013.02.040, 2013.

Laws, E. A.: Evaluation of In Situ Phytoplankton Growth Rates: A Synthesis of Data from Varied Approaches, Annu. Rev. Mar. Sci., 5(1), 247–268, doi:10.1146/annurev-marine-121211-172258, 2013.

Litchman, E., Klausmeier, C. A., Schofield, O. M. and Falkowski, P. G.: The role

of functional traits and trade-offs in structuring phytoplankton communities: scaling from cellular to ecosystem level, Ecol. Lett., 10(12), 1170–1181, doi:10.1111/j.1461-0248.2007.01117.x, 2007.

Marañón, E., Cermeño, P., López-Sandoval, D. C., Rodríguez-Ramos, T., Sobrino, C., Huete-Ortega, M., Blanco, J. M. and Rodríguez, J.: Unimodal size scaling of phytoplankton growth and the size dependence of nutrient uptake and use, Ecol. Lett., 16(3), 371–379, doi:10.1111/ele.12052, 2013.

Okada, H. and Honjo, S.: The distribution of oceanic coccolithophorids in the Pacific, Deep Sea Res. Oceanogr. Abstr., 20(4), 355–374, doi:10.1016/0011-7471(73)90059-4, 1973.

Rokitta, S. D. and Rost, B.: Effects of CO2 and their modulation by light in the life-cycle stages of the coccolithophore Emiliania huxleyi, Limnol. Oceanogr., 57(2), 607–618, doi:10.4319/lo.2012.57.2.0607, 2012.

Selph, K. E., Landry, M. R., Taylor, A. G., Yang, E.-J., Measures, C. I., Yang, J., Stukel, M. R., Christensen, S. and Bidigare, R. R.: Spatially-resolved taxon-specific phytoplankton production and grazing dynamics in relation to iron distributions in the Equatorial Pacific between 110 and 140°W, Deep Sea Res. Part II Top. Stud. Oceanogr., 58(3–4), 358–377, doi:10.1016/j.dsr2.2010.08.014, 2011.

Trimborn, S., Langer, G. and Rost, B.: Effect of varying calcium concentrations and light intensities on calcification and photosynthesis in Emiliania huxleyi, Limnol. Oceanogr., 52(5), 2285–2293, doi:10.4319/lo.2007.52.5.2285, 2007.

Zondervan, I., Rost, B. and Riebesell, U.: Effect of CO2 concentration on the PIC/POC ratio in the coccolithophore Emiliania huxleyi grown under light-limiting conditions and different daylengths, J. Exp. Mar. Biol. Ecol., 272(1), 55–70, doi:10.1016/S0022-0981(02)00037-0, 2002.

[Figure]

**Fig. 1.** L :D cycle calculated for the GYR station at the sampling day.

---

## Author Response (AR1)

**Author's response to comments to "Growth of the coccolithophore *Emiliania huxleyi* in light- and nutrient-limited batch reactors: relevance for the BIOSOPE deep ecological niche of coccolithophores", submitted by L. Perrin to Biogeosciences**

We have considered the comments made by reviewers 1 and 2 to our manuscript and the latter was modified as recommended. In addition to these corrections we modified text and repetitions and improved the quality of the scientific message overall the manuscript. Figures and tables as well were improved and changed when the results were not clear enough.

We found the reviewers' comments pertinent and think they improved the manuscript. We have included his suggestions in the revised manuscript. In the following we detail our responses to specific questions and are prepared to implement these corrections/changes should the article be accepted for publication in Biogeosciences.

**General comments from referees and author's response**

**Reviewer 1**

- The reviewer 1 suggests to merge certain figures and to place certain figures or tables in a supplementary material.

Figures 1, 8, 10, 11, 12 and table 3 were placed in a supplementary material. Figures 5 and 6 were merged such as figures 13 and 14. Table 5 was removed and values were described in the text of the manuscript.

**- The reviewer suggests applying the model to other literature data.**

We chose not to include other datasets in the present manuscript because this would have significantly increased the length of the manuscript without adding much in terms of the new modeling method proposed. We think that the modeling we present is strong enough to support our conclusions on the environmental controls on *E. huxleyi* distribution in the deep ecological niche of South Pacific Gyre. We hope our approach will also be used with other datasets in the future by other authors.

**- The reviewer points out the language of the manuscript and the long sentences.**

We improved considerably the language and make the text more concise avoiding repetition and long sentences. The subheadings of sub-sections 4.1.1 to 4.1.3 were deleted but the text was not deleted. The sub-sections were merged and the text was considerably reduced in order to be a short summary instead of only repetition.

**Reviewer 2**

- The reviewer 2 states that information on *Emiliania huxleyi* is not equal information on coccolithophores as a whole and suggests being more specific when discussing coccolithophores as a group or *E. huxleyi* as a single species.

We were really careful about this point throughout the manuscript and specified the species *E. huxleyi* when the "coccolithophores" term was not appropriate.

- The reviewer points out the focus on the deep niche and that wider implications of the study are potentially important.

The deep niche focus was chosen for two reasons: (1) little is known about *E. huxleyi* growth in these low-nutrient, low-light conditions despite the fact that they could represent a non-negligible portion of the global *E. huxleyi* population, and; (2) the BIOSOPE transect is unique in the breadth of physical and chemical parameters measured, which makes our joint experimental/modeling exercise easier. However, wider implications of the study for general oligotrophic regions than the deep niche of the South Pacific Gyre was taken into account in conclusions of the work:

"There is potential for our approach to shed light on the functioning of other oligotrophic, low-light phytoplankton ecosystems like cold, dark and nutrient-poor Arctic and Antarctic waters."

**- The reviewer states that the main message from this work is not clear enough and that figures and tables in the manuscript need to be merged, deleted or placed in Supplementary material.**

The main message of our work is that batch experiments coupled to simple physiological modeling can help interpret environmental controls on distributions of coccolithophore populations in the ocean. This message was delivered more clearly than in the original manuscript. The deep niche study was chosen to apply our approach is the best possible field situation based on the available published datasets of chemical and physical properties. Figures and tables were reorganized as: figures 1, 8, 10, 11, 12 and table 3 were placed in a supplementary material; figures 5 and 6 were merged such as figures 13 and 14; table 5 was removed and values were described in the text of the manuscript.

**- The reviewer suggests using the light dose as a comparison between different experimental studies rather than the amount of light.**

This point and the specification of the L:D cycle for each studies taken from the literature was added to the text and discussed in addition to the intensity of irradiance.

**- The reviewer points out that organic source of nitrogen could be use by *E. huxleyi* especially in oligotrophic environment.**

We added the following text: "A potential influence of organic nitrogen sources, that *E. huxleyi* is capable of using (Benner and Passow, 2010), cannot be excluded, but these would be expected to have been distributed vertically in a similar way to NO3."

**Specific comments and author's changes in manuscript Reviewer 1**

1) The period between units were removed in the manuscript through the text and in figures and tables.

2) Consistent color and marker were used in all figures to be clearer.

3) Line 18: The expression 'coccolithophore ecosystem' was not appropriate here and was changed to "potentially important ecological niche for coccolithophores".

4) L. 22-24: The word "physiology" was changed in "growth".

5) L. 40-41 vs L. 44-45: The sentence "Together, these effects modulate the impact of coccolithophores on ocean-atmosphere  $CO_2$  fluxes" in Ln. 44-45 was removed because of the repetition with the Ln. 40-41.

6) L. 113: We specified that we only added nitrate and phosphate to the medium and that we did not add the NH4Cl indicated in the reference medium in order to avoid the problem of multiple nitrogen sources.

7) L. 115: This sentence was changed to "Cells were acclimated to light, temperature and nutrient conditions for at least three growth cycles prior to experiments."

8) L. 120: Light intensity was expressed in  $\mu$ mol photons m-2 s-1.

9) L. 140: Samples were always taken in the afternoon between noon and 4pm, and always in the same sampling order. Each culture bottle sampling operation took approximately 45 min, preventing samples from different bottles from being taken at the same time. However, each analytic value was averaged over the three replicates. This was specified in the manuscript.

10) L. 149: This part of the sentence about coccolith width measurements was deleted.

11) L. 151: We mentioned that the error for pH measurements is 0.02 pH units.

12) L. 167: We added "PIC was obtained considering a 1:1 stoichiometry between Ca2+ and PIC, i.e. all of the calcium on the filters was considered to have come from calcium carbonate (Fagerbakke et al., 1994)."

13) L. 193: We made a mistake on this point. The C/N ratio for the nitrate uptake calculation is not necessary for this calculation because the PON data are available for the control experiments. To correct this point we changed the Monod plot (Fig. 5) and the part in the text that describes this point. This entailed only minor difference in the model results because the C/N ratios for the control experiments were near the Redfield ratio: for example the C/N ratio for the control NO3 experiment of Langer et al. (2013) was 5.72.

14) L. 199: To clarify notation, we used  $K_R$  for nutrients in general,  $K_N$  for the nitrate half-saturation constant and  $K_P$  for the phosphate half-saturation constant.

15) L. 210: The " $Q_N^{min}$ " was changed in " $Q_N^{max}$ ".

16) L. 277: This point was discussed in the discussion part of the manuscript: "The stationary phase was not attained in the P-limited low light culture, but it can be inferred that cells were P-limited from: (a) the POP quota, which was lower than that of the control, (b) the POC:POP ratio, which was higher than that of the control, and (c) a deviation of the growth curve from exponential growth starting (at the latest) on day 16 of 19. While a decline in POP quota is an early sign of limitation, the decline in growth rate occurs later, indicating more severe limitation. The cessation of cell division (stationary phase) would be the last stage in the process of becoming fully P-limited over the course of a batch culture".

17) L. 294: We made a mistake in calculating the magnitude of the error bar which is in fact smaller that we thought; thus we removed this sentence.

18) L. 379: The part of the sentence was changed to "numerically dominant coccolithophore E. huxleyi".

19) L. 381-382: We mentioned the Leonardos and Geider (2005) experiment because it is the only experiment to our knowledge where nutrient-and light co-limitation was carried out. We mentioned that this experiment was carried out with a non-calcifying strain.

20) L. 387: The sentence was changed in "...to ensure that changes in the carbonate system were within a minimal range (< 10% except for the P-limited experiment where the DIC change were 12 and 13%; Table 1)".

21) L. 396-399: This sentence was rephrased according to comment 16.

22) L. 452-458: This sentence was splited in several sentences.

23) L. 469: We removed "as well" in the sentence.

24) L. 527-529: This part was deleted to make shorter this part of the discussion.

25) L. 529-531: This part was deleted to make shorter this part of the discussion.

26) L. 550-554: A part of this sentence was deleted. The other part of the sentence was modified as "Claustre et al. (2008) reported a nitrate concentration <3 nM (i.e. below the detection limit) in the 0-100 m water column, whereas phosphate concentration was always above 0.1  $\mu$ M in surface layers (Raimbault and Garcia, 2008). Moutin et al. (2008) concluded that phosphate was apparently not the limiting nutrient for phytoplankton along the BIOSOPE transect".

27) L. 557-559: The sentence was changed in "Nitrification and the vertical diffusivity of nitrate through the nitracline (Holligan et al., 1984) needs to be taken into account and could potentially be a source of dissolved nitrate in the deep niche of coccolithophores."

28) L. 572-574: A sentence was added to mention the grazing and vertical export: "The maximum estimated growth rate at the GYR station (0.024 d-1 at 175 m depth) corresponds to an *E. huxleyi* generation time of 29.3 days, suggesting that division rate at the DCM was extremely slow, all the more so since this estimate does not consider grazing and vertical export of cells.".

29) L. 585-590: The sentence in question was splited into several sentences.

30) L. 610-614: This sentence was modified. While it is not possible to obtain reliable half-saturation constants for nutrient uptake in a batch experiment (a chemostat experiment is necessary), other parameters such as the maximum growth rates and maximum uptake rates can indeed be estimated in a batch experiment. As far as we know the only literature found to estimate the half-saturation constant for nutrient uptake for *E. huxleyi* using a batch culture is from Eppley et al. (1969). However, we think that the transient character of batch cultures makes the determination of half-saturation constants very difficult. We propose to circumvent this difficulty by modeling the batch experiments with a simple Droop model that enables us to extract information on nutrient affinity (the half saturation constant) from the transient results of the batch experiment

31) L. 679: Reference to the final revised version of Beaufort et al. (2008) was made.

32) Table 2: POC: PON and POC:POP was reported rather than PON:POC and POP:POC and the decimal point was used instead of the comma in the final manuscript (Table 2).

**Reviewer 2**

Ln 18: The expression 'coccolithophore ecosystem' was not appropriate here and was changed to "potentially important ecological niche for coccolithophores". The sentence was modified as "Alongside the well-known, shallow-water coccolithophore blooms visible from satellites, the lower photic zone is a poorly known but potentially important ecological niche for coccolithophores in terms of primary production and carbon export to deep ocean".

Ln 18-19: We changed the sentence as follow : "In this study, the physiological responses of an *Emiliania huxleyi* strain to conditions simulating the deep niche in the oligotrophic gyres along the BIOSOPE transect in the South Pacific oceanic gyre were investigated".

Ln 30: This sentence was modified to "This study contributes more widely to the understanding of *E. huxleyi* physiology and behavior in a low-light and oligotrophic environment of the ocean."

Lns 38-39: The word "contribute" was used rather than "participate".

Ln 40-41: The reviewer did a correct comment here and we modified the sentence as: "The relative importance of calcification and photosynthesis is one of the factors that dictates the effect of coccolithophores on ocean-atmosphere CO2 fluxes (Shutler et al., 2013). Environmental conditions such as temperature, irradiance, nutrient concentrations and pCO2 exert a primary control on the calcification/photosynthesis ratio in coccolithophores and also affect cellular growth rates, which, together with grazing, mortality, sinking of cells and oceanic transport, define the biogeography of coccolithophores.".

Ln 42-43: We added "in coccolithophores" to avoid confusion with the whole phytoplankton community.

Ln 44: As detailed in the comments "Ln 40-41", the sentence was changed.

Ln 47: We started the list with "e.g." as well in the Ln 50.

Ln 60: The term "discovered" was changed to "observed". Deep photic zone (low light) communities of coccolithophores have been observed in the North and Central Pacific at least since the work of Okada and Honjo (1973).

Ln 62: The sentence was modified as "This deep coccolithophore niche occurred at about 200 m depth, at a very low irradiance level (< 20  $\mu$ mol photons m-2 s-1) and at a depth corresponding to the nitrate and phosphate nutricline with dissolved nitrate (NO3) and phosphate (PO4) concentrations of about 1  $\mu$ M and 0.2  $\mu$ M, respectively."

Ln 114: We chose to work with a surface strain from the BIOSOPE transect because no *E. huxleyi* strains were isolated inside the gyre at 200 m depth. This is a limitation of our study that we will mention.

Lns 120-123: A model of the PAR daily cycle at the date and the coordinates of the GYR station was used to calculate the L:D cycle. This was between 14:10 and 12:12 along the whole transect. Thus, the 12:12 cycle used in our experiments is representative of the in situ situation. This point was specified in the manuscript: "taken from a calculation of L:D cycle at the GYR station at the date of the sampling".

Ln 159: Samples for nutrients were analyzed on a Seal Analytical auto-analyzer model AA3. This was modified in the text of the manuscript.

http://www.seal-analytical.com/Products/AA3HRAutoAnalyzer/tabid/59/language/en-US/Default.aspx.

Ln 168-170: The details were added to the manuscript: "POP was determined as the difference between the total particulate phosphorus and particulate inorganic phosphorus, analyzed using a auto-analyser Seal Analytical AA3, after the filters were placed in a solution of hydrochloric acid, according to the method of Labry et al. (2013).".

Ln 189: It is N-uptake. This was changed in the text.

Ln 192-194: We made a mistake on this point. The C/N ratio for the nitrate uptake calculation is not necessary for this calculation because the PON data are available for the control experiments. To correct this point we will change the Monod plot (Fig. 5 in the manuscript) and the text that describes this point. This will entail only a minor difference in the model results because the C/N ratios for the control experiment of Langer et al. (2013) was 5.72.

Ln 202-205: We improved the sentence about these two different methods to determine cell volume and surface area: "The volume and surface of cells ( $S_{cell}$ ) was obtained either by measurements of cells (both in the control culture and at the end of the nutrient-limited cultures) for the RCC911 strain experiments, or was estimated from  $Q_c$  the cellular organic carbon quota (in pmolc cell-1), and the density of carbon in coccolithophore biomass (approximately equal to 0.015 pmolc  $\mu$ m-3; Aloisi, 2015) for the batch experiments of Langer et al. (2013) for which cell measurements were not made",

Ln 214: We changed "NO $_3$  and PO $_4$ " to "N and P" to avoid confusion.

Ln 216: We changed the nutrients notation in the text because of the existing confusion between nutrient N and nitrogen N. We referred to nutrients in general with the letter R, to the nutrient nitrogen with the letter N and to nutrient phosphate with the letter P.

Ln 243: We changed the notation for the half saturation constants for nutrient uptake:  $K_N$  is the constant for nitrate uptake,  $K_P$  is the constant for phosphate uptake and  $K_R$  is the generalized constant for nutrient uptake. Same thing for the nutrient quotas, e.g.  $Q_{N/P}$ , that was referred to as  $Q_R$ .

Ln 250-251: We mean nutrient cellular quota and we added this point to the text.

Ln 269,279,288,306: We removed these sub-headings.

Ln 293: We expressed ratios as C:P and C:N when revising the manuscript rather than P:C and N:C.

Ln 379-381: We were more careful when we talk about *E. huxleyi* and coccolithophores as a group. Of course this work gives us new insights for the species *E. huxleyi* and maybe for other Isochrysidales or Noelaerhabdaceae but undoubtedly not for all coccolithophore species.

Ln 442: We changed "for decreasing phosphate than for decreasing nitrate" to " for P- limitation than for Nlimitation".

Ln 447: We modified this sentence making it clear that Zondervan (2007) is almost entirely based on *E. huxleyi* results.

Ln 452-460: As in general comment, the light dose was added to the text in order to improve the comparison and because of the importance of the light dose and not only the light intensity. Only Feng et al. (2008) used a 12:12 L:D cycle, but the other mentioned studies Rokitta and Rost (2012), Trimborn et al., (2007) and Zondervan et al. (2002) used a 16:8 L:D cycle. We changed this paragraph to be more specific and avoid comparing experiments with very different L:D cycle experiments.

Ln 463-465: As noted by the reviewer the relationship between coccosphere size and coccolith size is very species-specific, thus we decided to remove a part of this sentence and modified it as "The significant correlation between cell and coccosphere volume (Figure 4) and observations of other studies (e.g. Aloisi, 2015; Gibbs et al., 2013) support the conclusion that coccosphere size in the water column and in sediments could be used as a proxy for cell size (and thus POC quota).".

Ln 467,476, 483: These summary sections were combined and written more clearly.

Ln 561: This sentence was rephrased. Other sources of nitrogen might include organic nitrogen, although based on the modeling results (see answer to general comment) we think that inorganic nitrogen dominates over organic nitrogen.

"As *E. huxleyi* is capable to use organic sources of nitrogen as shown by Benner and Passow (2010), this nitrogen source cannot be excluded, but these would be expected to have been distributed vertically in a similar way to  $NO_3$ ".

[revised manuscript text omitted]

Interligne : Multiple 1,15 li                           | pt,       |
|--------|-----------------------------------------------------------------------------------------------------------------------------------|-------------------|---------------------------------------------------------------------------------------------|-----------|
| 1      | Growth of the coccolithophore Emiliania huxleyi in light- and nutrient-limited batch                                              |                   | Mis en forme : Numérotation :
Recommencer à chaque section                               |           |
| 2      | reactors: relevance for the BIOSOPE deep ecological niche of coccolithophores                                                     |                   | Mis en forme : Police :+Corps, 12                                                           | pt        |
| 4      | Laura Perrin + , Jan Probert 2 , Gerald Langer 3 and Giovanni Aloisi 4                | _                 | Mis en forme : Police : (Par défaut                                                         | :)        |
| 5      | 1 Sorbonne Universités, UPMC Univ. Paris 06 -CNRS-IRD-MNHN, LOCEAN-IPSL, 75252 Paris, France,                          | $\setminus$       | +Corps, 11 pt, Français (France), N
Étendu de/ Condensé de                               | on        |
| 6      | 2 CNRS-UPMC Univ. Paris 06 FR2424, Roscoff Culture Collection, Station Biologique de Roscoff, 29680 Roscoff, France.   | Ζ,                | Supprimé: Ian Probert 2 , Gerald                                                 | I         |
| 7      | 3 The Marine Biological Association of the United Kingdom, The Laboratory, Citadel Hill, Plymouth, Devon, PL1 2PB, UK, | $\backslash$      | Mis en forme                                                                                | (         |
| 8
9 | 4 LOCEAN, UMR 7159, CNRS-UPMC-IRD-MNHN, 75252 Paris, France.                                                           |                   | Mis en forme : Police :(Par défaut
+Corps, Français (France), Non Éte
de/ Condensé de | :)
ndu |
| 10     | Correspondence to: L. Perrin ( lpelod@locean-ipsl.upmc.fr )                                                                | ()()              | Supprimé:                                                                                   |           |
| 12     | Abstract, Coccolithophores are unicellular calcifying marine algae that play an important role in the 👘 🔸                         | (  )              | Supprimé: 3 Marine                                                               |           |
| 13     | oceanic carbon cycle via their cellular processes of photosynthesis (a $CO_2$ sink) and calcification (a $CO_2$                   |                   | Mis en forme                                                                                | (  |
| 1/     | source) In contrast to the well-studied, shallow-water coccolithonhore blooms visible from satellites, the                        |                   | His en forme : Police :(Par defaut
+Corps, Non Étendu de/ Condensé                       | :)
de  |
| 14     | source). In contrast to the weil studied, shallow-water coccontribution booths visible from satellites, the                       | $\left( \right) $ | Supprimé:                                                                                   |           |
| 15     | lower photic zone is a poorly known but potentially important ecological niche for coccolithophores in                            |                   | Mis en forme : Police : (Par défaut                                                         | :)
do  |
| 16     | terms of primary production and carbon export to the deep ocean. In this study, the physiological                                 |                   | Supprimé: 1                                                                                 | ue        |
| 17     | responses of an Emiliania huxleyi strain to conditions simulating the deep niche in the oligotrophic gyres                        |                   | Mis en forme                                                                                |           |
| 18     | along the BIOSOPE transect in the South Pacific oceanic gyre were investigated. We carried out batch                              |                   | Code de champ modifié                                                                       |           |
| 19     | culture experiments with an E. huxleyi strain isolated from the BIOSOPE transect, reproducing the in situ                  |                   | Mis en forme                                                                                |    |
| 20     | conditions of light- and nutrient- (nitrate and phosphate) limitation. By simulating coccolithophore growth                       |                   | Mis en forme : Non souligné, Coul
de police : Automatique                                | leur      |
| 21     | using an internal stores (Droop) model, we were able to constrain fundamental physiological parameters                            |                   | Mis en forme : Espace Avant : 0 p
Ne pas ajuster l'espace entre le text                  | ot,
te |
| 22     | for this E. huxleyi strain. We show that simple batch experiments, in conjunction with physiological                |                   | latin et asiatique, Ne pas ajuster
l'espace entre le texte et les nombre                 | es        |
| 23     | modelling, can provide reliable estimates of fundamental physiological parameters for E. huxleyi that are                  |                   | asiatiques                                                                                  |           |
| 24     | usually obtained experimentally in more time-consuming and costly chemostat experiments. The                                      |                   | Supprimé: ¶                                                                                 |           |
| 25     | combination of culture experiments, physiological modelling and in situ data from the PLOSOPE cruice                              |                   | Mis en forme                                                                                | (  |
| 25     | combination of culture experiments, physiological modeling and in situ data from the biosofic cluse                               |                   | Mis en forme : Non souligné Cou                                                             | leur      |
| 26     | shows that E. huxley growth in the deep BIOSOPE niche is co-limited by availability of light and nitrate. This             |                   | de police : Automatique                                                                     |           |
| 27     | study contributes more widely to the understanding of E. huxleyi physiology and behavior in a Jow-light and         |                   | Supprimé: known                                                                             |           |
| 28     | oligotrophic environment of the ocean.                                                                                            |                   | Mis en forme : Non souligné, Coul
de police : Automatique                                | leur      |
| 29     | Keywords                                                                                                                          |                   | Mis en forme                                                                                | (         |
| 30     | Coccolithophores, batch cultures, deep niche, South Pacific Gyre, Droop model , physiological parameters .          |                   | Supprime: we
|        |                                                                                                                                   |                   | I'IIS CII IUIIIIC                                                                           | 1         |

32

33

34

35

10

...

...

...

**1. Introduction**

[revised manuscript text omitted]

de police : Automatique                          | eur      |
|---------------------------------------------------------------------------------------|----------|
de police : Automatique                          | eur      |
de police : Automatique                          | eur      |
de police : Automatique                          | eur      |
Microscope.                                            |          |
de police : Automatique, Anglais
(États-Unis) | eur      |
|                                                                                       | 1        |

[revised manuscript text omitted]

|       |                                                                                                                                                                                                                                                                                                                                                                                                                                                                                                                                                                                                                                                                                                                                                                                                                                                                                                                                                                                                                                                                                                                                                                                                                                                                                                                                                                                                                                                                                                                                                                                                                                                                                                                                                                                                                                                                                                                                                                                                                                                                                                                                                                                                                                                                                                                                                                                                                                                                                                                                                                                                                                                                                                                                                                                                                                                                                                                                                                                    | ,                                                                                                                                                                                                                                                                                                                                                                                                                                                                                                                                                                                                                                                                                                                                                                                                                                                                                                                                                                                                                                                                                                                                                                                                                                                                                                                                                                                                                                                                                                                                                                                                                                                                                                                                                                                                                                                                                                                                                                                                                                                                                                                              | Su        | pprimé: $N_{up} = S_{cell} \cdot V_{m}$          |                  |
|-------|------------------------------------------------------------------------------------------------------------------------------------------------------------------------------------------------------------------------------------------------------------------------------------------------------------------------------------------------------------------------------------------------------------------------------------------------------------------------------------------------------------------------------------------------------------------------------------------------------------------------------------------------------------------------------------------------------------------------------------------------------------------------------------------------------------------------------------------------------------------------------------------------------------------------------------------------------------------------------------------------------------------------------------------------------------------------------------------------------------------------------------------------------------------------------------------------------------------------------------------------------------------------------------------------------------------------------------------------------------------------------------------------------------------------------------------------------------------------------------------------------------------------------------------------------------------------------------------------------------------------------------------------------------------------------------------------------------------------------------------------------------------------------------------------------------------------------------------------------------------------------------------------------------------------------------------------------------------------------------------------------------------------------------------------------------------------------------------------------------------------------------------------------------------------------------------------------------------------------------------------------------------------------------------------------------------------------------------------------------------------------------------------------------------------------------------------------------------------------------------------------------------------------------------------------------------------------------------------------------------------------------------------------------------------------------------------------------------------------------------------------------------------------------------------------------------------------------------------------------------------------------------------------------------------------------------------------------------------------------|--------------------------------------------------------------------------------------------------------------------------------------------------------------------------------------------------------------------------------------------------------------------------------------------------------------------------------------------------------------------------------------------------------------------------------------------------------------------------------------------------------------------------------------------------------------------------------------------------------------------------------------------------------------------------------------------------------------------------------------------------------------------------------------------------------------------------------------------------------------------------------------------------------------------------------------------------------------------------------------------------------------------------------------------------------------------------------------------------------------------------------------------------------------------------------------------------------------------------------------------------------------------------------------------------------------------------------------------------------------------------------------------------------------------------------------------------------------------------------------------------------------------------------------------------------------------------------------------------------------------------------------------------------------------------------------------------------------------------------------------------------------------------------------------------------------------------------------------------------------------------------------------------------------------------------------------------------------------------------------------------------------------------------------------------------------------------------------------------------------------------------|-----------|--------------------------------------------------|------------------|
|       |                                                                                                                                                                                                                                                                                                                                                                                                                                                                                                                                                                                                                                                                                                                                                                                                                                                                                                                                                                                                                                                                                                                                                                                                                                                                                                                                                                                                                                                                                                                                                                                                                                                                                                                                                                                                                                                                                                                                                                                                                                                                                                                                                                                                                                                                                                                                                                                                                                                                                                                                                                                                                                                                                                                                                                                                                                                                                                                                                                                    |                                                                                                                                                                                                                                                                                                                                                                                                                                                                                                                                                                                                                                                                                                                                                                                                                                                                                                                                                                                                                                                                                                                                                                                                                                                                                                                                                                                                                                                                                                                                                                                                                                                                                                                                                                                                                                                                                                                                                                                                                                                                                                                                | Mis       | s en forme                                       |                  |
|       |                                                                                                                                                                                                                                                                                                                                                                                                                                                                                                                                                                                                                                                                                                                                                                                                                                                                                                                                                                                                                                                                                                                                                                                                                                                                                                                                                                                                                                                                                                                                                                                                                                                                                                                                                                                                                                                                                                                                                                                                                                                                                                                                                                                                                                                                                                                                                                                                                                                                                                                                                                                                                                                                                                                                                                                                                                                                                                                                                                                    |                                                                                                                                                                                                                                                                                                                                                                                                                                                                                                                                                                                                                                                                                                                                                                                                                                                                                                                                                                                                                                                                                                                                                                                                                                                                                                                                                                                                                                                                                                                                                                                                                                                                                                                                                                                                                                                                                                                                                                                                                                                                                                                                | Su        | oprimé: 🛛                                        |                  |
| 536   | $R_{up} = S_{cell} \cdot V_{\max R} \cdot \frac{[\Gamma]}{[R] + K_p} $ (5)                                                                                                                                                                                                                                                                                                                                                                                                                                                                                                                                                                                                                                                                                                                                                                                                                                                                                                                                                                                                                                                                                                                                                                                                                                                                                                                                                                                                                                                                                                                                                                                                                                                                                                                                                                                                                                                                                                                                                                                                                                                                                                                                                                                                                                                                                                                                                                                                                                                                                                                                                                                                                                                                                                                                                                                                                                                                                                         |                                                                                                                                                                                                                                                                                                                                                                                                                                                                                                                                                                                                                                                                                                                                                                                                                                                                                                                                                                                                                                                                                                                                                                                                                                                                                                                                                                                                                                                                                                                                                                                                                                                                                                                                                                                                                                                                                                                                                                                                                                                                                                                                | Suj       | pprimé: V max                         | $\rightarrow$    |
|       |                                                                                                                                                                                                                                                                                                                                                                                                                                                                                                                                                                                                                                                                                                                                                                                                                                                                                                                                                                                                                                                                                                                                                                                                                                                                                                                                                                                                                                                                                                                                                                                                                                                                                                                                                                                                                                                                                                                                                                                                                                                                                                                                                                                                                                                                                                                                                                                                                                                                                                                                                                                                                                                                                                                                                                                                                                                                                                                                                                                    |                                                                                                                                                                                                                                                                                                                                                                                                                                                                                                                                                                                                                                                                                                                                                                                                                                                                                                                                                                                                                                                                                                                                                                                                                                                                                                                                                                                                                                                                                                                                                                                                                                                                                                                                                                                                                                                                                                                                                                                                                                                                                                                                | Suj       | oprimé: 🛛                                        | —                |
| F 2 7 |                                                                                                                                                                                                                                                                                                                                                                                                                                                                                                                                                                                                                                                                                                                                                                                                                                                                                                                                                                                                                                                                                                                                                                                                                                                                                                                                                                                                                                                                                                                                                                                                                                                                                                                                                                                                                                                                                                                                                                                                                                                                                                                                                                                                                                                                                                                                                                                                                                                                                                                                                                                                                                                                                                                                                                                                                                                                                                                                                                                    |                                                                                                                                                                                                                                                                                                                                                                                                                                                                                                                                                                                                                                                                                                                                                                                                                                                                                                                                                                                                                                                                                                                                                                                                                                                                                                                                                                                                                                                                                                                                                                                                                                                                                                                                                                                                                                                                                                                                                                                                                                                                                                                                | Suj       | pprimé: mol N .ℤ                      |                  |
| 537   | where $S_{Cell}$ (in $\mu$ m) is the surface area of the cell, $V_{maxR}$ (in $\mu$ mol R $\mu$ m $_{2}$ d ) is the maximum surface-                                                                                                                                                                                                                                                                                                                                                                                                                                                                                                                                                                                                                                                                                                                                                                                                                                                                                                                                                                                                                                                                                                                                                                                                                                                                                                                                                                                                                                                                                                                                                                                                                                                                                                                                                                                                                                                                                                                                                                                                                                                                                                                                                                                                                                                                                                                                                                                                                                                                                                                                                                                                                                                                                                                                                                                                                                    | $\leftarrow$                                                                                                                                                                                                                                                                                                                                                                                                                                                                                                                                                                                                                                                                                                                                                                                                                                                                                                                                                                                                                                                                                                                                                                                                                                                                                                                                                                                                                                                                                                                                                                                                                                                                                                                                                                                                                                                                                                                                                                                                                                                                                                                   | Suj       | pprimé: .                                        |                  |
| 538   | normalized nutrient uptake rate (obtained by fitting the model to the data) and $\underline{K}_{R}$ (in $\mu$ mol $L^{-1}$ ) is the                                                                                                                                                                                                                                                                                                                                                                                                                                                                                                                                                                                                                                                                                                                                                                                                                                                                                                                                                                                                                                                                                                                                                                                                                                                                                                                                                                                                                                                                                                                                                                                                                                                                                                                                                                                                                                                                                                                                                                                                                                                                                                                                                                                                                                                                                                                                                                                                                                                                                                                                                                                                                                                                                                                                                                                                                                                |                                                                                                                                                                                                                                                                                                                                                                                                                                                                                                                                                                                                                                                                                                                                                                                                                                                                                                                                                                                                                                                                                                                                                                                                                                                                                                                                                                                                                                                                                                                                                                                                                                                                                                                                                                                                                                                                                                                                                                                                                                                                                                                                | Mis       | s en forme                                       |                  |
| 539   | (Michaelis-Menten) half-saturation constant for uptake of nutrient $\underline{R}$ . The volume and surface of cells ( $S_{cell}$ )                                                                                                                                                                                                                                                                                                                                                                                                                                                                                                                                                                                                                                                                                                                                                                                                                                                                                                                                                                                                                                                                                                                                                                                                                                                                                                                                                                                                                                                                                                                                                                                                                                                                                                                                                                                                                                                                                                                                                                                                                                                                                                                                                                                                                                                                                                                                                                                                                                                                                                                                                                                                                                                                                                                                                                                                                                                |                                                                                                                                                                                                                                                                                                                                                                                                                                                                                                                                                                                                                                                                                                                                                                                                                                                                                                                                                                                                                                                                                                                                                                                                                                                                                                                                                                                                                                                                                                                                                                                                                                                                                                                                                                                                                                                                                                                                                                                                                                                                                                                                | Mis       | s en forme                                       |                  |
| 540   | was obtained either by measurements of cells (both in the control culture and at the end of the nutrient-                                                                                                                                                                                                                                                                                                                                                                                                                                                                                                                                                                                                                                                                                                                                                                                                                                                                                                                                                                                                                                                                                                                                                                                                                                                                                                                                                                                                                                                                                                                                                                                                                                                                                                                                                                                                                                                                                                                                                                                                                                                                                                                                                                                                                                                                                                                                                                                                                                                                                                                                                                                                                                                                                                                                                                                                                                                                   |                                                                                                                                                                                                                                                                                                                                                                                                                                                                                                                                                                                                                                                                                                                                                                                                                                                                                                                                                                                                                                                                                                                                                                                                                                                                                                                                                                                                                                                                                                                                                                                                                                                                                                                                                                                                                                                                                                                                                                                                                                                                                                                                | Mis       | s en forme                                       |                  |
| 541   | limited cultures) for the RCC911 strain experiments, or was estimated from O c , the cellular organic carbon                                                                                                                                                                                                                                                                                                                                                                                                                                                                                                                                                                                                                                                                                                                                                                                                                                                                                                                                                                                                                                                                                                                                                                                                                                                                                                                                                                                                                                                                                                                                                                                                                                                                                                                                                                                                                                                                                                                                                                                                                                                                                                                                                                                                                                                                                                                                                                                                                                                                                                                                                                                                                                                                                                                                                                                                                                                            | M                                                                                                                                                                                                                                                                                                                                                                                                                                                                                                                                                                                                                                                                                                                                                                                                                                                                                                                                                                                                                                                                                                                                                                                                                                                                                                                                                                                                                                                                                                                                                                                                                                                                                                                                                                                                                                                                                                                                                                                                                                                                                                                              | Mis       | s en forme                                       |                  |
| 511   | $\frac{1}{1000}$ and $\frac{1}{1000}$ and the density of earlier is essentiated from $\frac{1}{2000}$ (are contained or gains and the 0.015                                                                                                                                                                                                                                                                                                                                                                                                                                                                                                                                                                                                                                                                                                                                                                                                                                                                                                                                                                                                                                                                                                                                                                                                                                                                                                                                                                                                                                                                                                                                                                                                                                                                                                                                                                                                                                                                                                                                                                                                                                                                                                                                                                                                                                                                                                                                                                                                                                                                                                                                                                                                                                                                                                                                                                                                                                        |                                                                                                                                                                                                                                                                                                                                                                                                                                                                                                                                                                                                                                                                                                                                                                                                                                                                                                                                                                                                                                                                                                                                                                                                                                                                                                                                                                                                                                                                                                                                                                                                                                                                                                                                                                                                                                                                                                                                                                                                                                                                                                                                | Mis       | s en forme                                       |                  |
| 542   | quota (in prior e ceir), and the density of carbon in coccolitrophore biomass (approximately equal to 0.015                                                                                                                                                                                                                                                                                                                                                                                                                                                                                                                                                                                                                                                                                                                                                                                                                                                                                                                                                                                                                                                                                                                                                                                                                                                                                                                                                                                                                                                                                                                                                                                                                                                                                                                                                                                                                                                                                                                                                                                                                                                                                                                                                                                                                                                                                                                                                                                                                                                                                                                                                                                                                                                                                                                                                                                                                                                             | $\langle \  \ $                                                                                                                                                                                                                                                                                                                                                                                                                                                                                                                                                                                                                                                                                                                                                                                                                                                                                                                                                                                                                                                                                                                                                                                                                                                                                                                                                                                                                                                                                                                                                                                                                                                                                                                                                                                                                                                                                                                                                                                                                                                                                                                | MIS
Su | onrimé: K (in D                                  |           |
| 543   | pmolc um -3 ; Aloisi, 2015) for the batch experiments of Langer et al. (2013) for which cell measurements                                                                                                                                                                                                                                                                                                                                                                                                                                                                                                                                                                                                                                                                                                                                                                                                                                                                                                                                                                                                                                                                                                                                                                                                                                                                                                                                                                                                                                                                                                                                                                                                                                                                                                                                                                                                                                                                                                                                                                                                                                                                                                                                                                                                                                                                                                                                                                                                                                                                                                                                                                                                                                                                                                                                                                                                                                                               |                                                                                                                                                                                                                                                                                                                                                                                                                                                                                                                                                                                                                                                                                                                                                                                                                                                                                                                                                                                                                                                                                                                                                                                                                                                                                                                                                                                                                                                                                                                                                                                                                                                                                                                                                                                                                                                                                                                                                                                                                                                                                                                                | Su        |                                                  |           |
| 544   | were not made .                                                                                                                                                                                                                                                                                                                                                                                                                                                                                                                                                                                                                                                                                                                                                                                                                                                                                                                                                                                                                                                                                                                                                                                                                                                                                                                                                                                                                                                                                                                                                                                                                                                                                                                                                                                                                                                                                                                                                                                                                                                                                                                                                                                                                                                                                                                                                                                                                                                                                                                                                                                                                                                                                                                                                                                                                                                                                                                                                             |                                                                                                                                                                                                                                                                                                                                                                                                                                                                                                                                                                                                                                                                                                                                                                                                                                                                                                                                                                                                                                                                                                                                                                                                                                                                                                                                                                                                                                                                                                                                                                                                                                                                                                                                                                                                                                                                                                                                                                                                                                                                                                                                | Mis       | sen forme                                        |                  |
| 545   | The phytoplankton growth rate $\mu$ (in d -1 ) was calculated based on the normalized n Quota equation reported                                                                                                                                                                                                                                                                                                                                                                                                                                                                                                                                                                                                                                                                                                                                                                                                                                                                                                                                                                                                                                                                                                                                                                                                                                                                                                                                                                                                                                                                                                                                                                                                                                                                                                                                                                                                                                                                                                                                                                                                                                                                                                                                                                                                                                                                                                                                                                                                                                                                                                                                                                                                                                                                                                                                                                                                                                              |                                                                                                                                                                                                                                                                                                                                                                                                                                                                                                                                                                                                                                                                                                                                                                                                                                                                                                                                                                                                                                                                                                                                                                                                                                                                                                                                                                                                                                                                                                                                                                                                                                                                                                                                                                                                                                                                                                                                                                                                                                                                                                                                | Mis       | s en forme                                       |                  |
| 546   | in Elvon (2008):                                                                                                                                                                                                                                                                                                                                                                                                                                                                                                                                                                                                                                                                                                                                                                                                                                                                                                                                                                                                                                                                                                                                                                                                                                                                                                                                                                                                                                                                                                                                                                                                                                                                                                                                                                                                                                                                                                                                                                                                                                                                                                                                                                                                                                                                                                                                                                                                                                                                                                                                                                                                                                                                                                                                                                                                                                                                                                                                                                   |                                                                                                                                                                                                                                                                                                                                                                                                                                                                                                                                                                                                                                                                                                                                                                                                                                                                                                                                                                                                                                                                                                                                                                                                                                                                                                                                                                                                                                                                                                                                                                                                                                                                                                                                                                                                                                                                                                                                                                                                                                                                                                                                | Suj       | pprimé: N The volume and s                       | surfa(           |
| 540   |                                                                                                                                                                                                                                                                                                                                                                                                                                                                                                                                                                                                                                                                                                                                                                                                                                                                                                                                                                                                                                                                                                                                                                                                                                                                                                                                                                                                                                                                                                                                                                                                                                                                                                                                                                                                                                                                                                                                                                                                                                                                                                                                                                                                                                                                                                                                                                                                                                                                                                                                                                                                                                                                                                                                                                                                                                                                                                                                                                                    |                                                                                                                                                                                                                                                                                                                                                                                                                                                                                                                                                                                                                                                                                                                                                                                                                                                                                                                                                                                                                                                                                                                                                                                                                                                                                                                                                                                                                                                                                                                                                                                                                                                                                                                                                                                                                                                                                                                                                                                                                                                                                                                                | Mis       | s en forme                                       |                  |
|       | $(1 + KO) \cdot (O - O^{\min})$                                                                                                                                                                                                                                                                                                                                                                                                                                                                                                                                                                                                                                                                                                                                                                                                                                                                                                                                                                                                                                                                                                                                                                                                                                                                                                                                                                                                                                                                                                                                                                                                                                                                                                                                                                                                                                                                                                                                                                                                                                                                                                                                                                                                                                                                                                                                                                                                                                                                                                                                                                                                                                                                                                                                                                                                                                                                                                                                                    |                                                                                                                                                                                                                                                                                                                                                                                                                                                                                                                                                                                                                                                                                                                                                                                                                                                                                                                                                                                                                                                                                                                                                                                                                                                                                                                                                                                                                                                                                                                                                                                                                                                                                                                                                                                                                                                                                                                                                                                                                                                                                                                                | Su        | pprimé: )                                        |                  |
| 547   | $\mu = \mu_{\max} \cdot \frac{(1 + KQ_R) \cdot (Q - Q_R)}{(Q - Q^{\min}) + KQ - (Q^{\max} - Q^{\min})} $ (6)                                                                                                                                                                                                                                                                                                                                                                                                                                                                                                                                                                                                                                                                                                                                                                                                                                                                                                                                                                                                                                                                                                                                                                                                                                                                                                                                                                                                                                                                                                                                                                                                                                                                                                                                                                                                                                                                                                                                                                                                                                                                                                                                                                                                                                                                                                                                                                                                                                                                                                                                                                                                                                                                                                                                                                                                                                                                       | And a state of the | Co        | de de champ modifié                              |                  |
|       | $(Q-Q_R) + KQ_R \cdot (Q_R - Q_R)$                                                                                                                                                                                                                                                                                                                                                                                                                                                                                                                                                                                                                                                                                                                                                                                                                                                                                                                                                                                                                                                                                                                                                                                                                                                                                                                                                                                                                                                                                                                                                                                                                                                                                                                                                                                                                                                                                                                                                                                                                                                                                                                                                                                                                                                                                                                                                                                                                                                                                                                                                                                                                                                                                                                                                                                                                                                                                                                                                 |                                                                                                                                                                                                                                                                                                                                                                                                                                                                                                                                                                                                                                                                                                                                                                                                                                                                                                                                                                                                                                                                                                                                                                                                                                                                                                                                                                                                                                                                                                                                                                                                                                                                                                                                                                                                                                                                                                                                                                                                                                                                                                                                | Mis       | s en forme                                       |                  |
| 548   | ــــــــــــــــــــــــــــــــــــ                                                                                                                                                                                                                                                                                                                                                                                                                                                                                                                                                                                                                                                                                                                                                                                                                                                                                                                                                                                                                                                                                                                                                                                                                                                                                                                                                                                                                                                                                                                                                                                                                                                                                                                                                                                                                                                                                                                                                                                                                                                                                                                                                                                                                                                                                                                                                                                                                                                                                                                                                                                                                                                                                                                                                                                                                                                                                                                                        |                                                                                                                                                                                                                                                                                                                                                                                                                                                                                                                                                                                                                                                                                                                                                                                                                                                                                                                                                                                                                                                                                                                                                                                                                                                                                                                                                                                                                                                                                                                                                                                                                                                                                                                                                                                                                                                                                                                                                                                                                                                                                                                                | Su        | oprimė: (Aloisi, 2015).                          |                  |
| 549   | where $\mu_{max}$ (in $\underline{d}^{-1}$ ) is the maximum growth rate attained at the maximum nutrient cell quota $\underline{Q}_{R}^{max}$ (in $\underline{\mu}$ mol                                                                                                                                                                                                                                                                                                                                                                                                                                                                                                                                                                                                                                                                                                                                                                                                                                                                                                                                                                                                                                                                                                                                                                                                                                                                                                                                                                                                                                                                                                                                                                                                                                                                                                                                                                                                                                                                                                                                                                                                                                                                                                                                                                                                                                                                                                                                                                                                                                                                                                                                                                                                                                                                                                                                                                                                            |                                                                                                                                                                                                                                                                                                                                                                                                                                                                                                                                                                                                                                                                                                                                                                                                                                                                                                                                                                                                                                                                                                                                                                                                                                                                                                                                                                                                                                                                                                                                                                                                                                                                                                                                                                                                                                                                                                                                                                                                                                                                                                                                | Mis       | s en forme                                       |                  |
| 550   | cell -1 ), $Q_{R}^{min}$ (in $\mu$ mol cell -1 ) is the minimum (subsistence) cellular quota of nutrient R below which growth                                                                                                                                                                                                                                                                                                                                                                                                                                                                                                                                                                                                                                                                                                                                                                                                                                                                                                                                                                                                                                                                                                                                                                                                                                                                                                                                                                                                                                                                                                                                                                                                                                                                                                                                                                                                                                                                                                                                                                                                                                                                                                                                                                                                                                                                                                                                                                                                                                                                                                                                                                                                                                                                                                                                                                                                                                |                                                                                                                                                                                                                                                                                                                                                                                                                                                                                                                                                                                                                                                                                                                                                                                                                                                                                                                                                                                                                                                                                                                                                                                                                                                                                                                                                                                                                                                                                                                                                                                                                                                                                                                                                                                                                                                                                                                                                                                                                                                                                                                                |           | le de chamn modifié                              |           |
| 551   | stops and $KQ_{g}$ is a dimensionless parameter that can be readily compared between nutrient types and                                                                                                                                                                                                                                                                                                                                                                                                                                                                                                                                                                                                                                                                                                                                                                                                                                                                                                                                                                                                                                                                                                                                                                                                                                                                                                                                                                                                                                                                                                                                                                                                                                                                                                                                                                                                                                                                                                                                                                                                                                                                                                                                                                                                                                                                                                                                                                                                                                                                                                                                                                                                                                                                                                                                                                                                                                                                            |